# A Review of Research on the Impact Mechanisms of Green Development in the Transportation Industry

Yumeng Mao and Xuemei Li *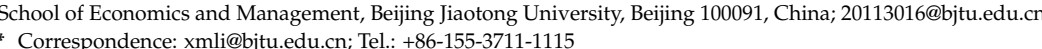

School of Economics and Management, Beijing Jiaotong University, Beijing 100091, China; 20113016@bjtu.edu.cn
* Correspondence: xmli@bjtu.edu.cn; Tel.: +86-155-3711-1115

**Abstract:** Green development in the transportation industry is a new type of development. As the huge energy consumption and carbon emissions generated by the transportation industry have caused many environmental problems, the healthy and environmentally friendly mode of industrial development has received more and more attention. However, the quantification of green development in the transportation industry varies in terms of boundaries, scope, and methods. Due to digital empowerment, the degree of influence and direction of the factors affecting the green development are not fixed. The prediction of future development prospects is relatively single-minded, lacking a comprehensive simulation scenario setting from multiple perspectives. This paper systematically reviews the research progress of green development of the transportation industry from three aspects: development performance assessment, influence mechanism analysis, and development path exploration. After a critical analysis, this study concludes that (1) a clear methodology is needed to assess the direct and indirect non-desired output results of the transportation industry; (2) considering the endogenization of the level of technology, the influence of the interaction between the influencing factors, etc., on the degree and direction of the role of the factors, a more scientific econometric model should be established for in-depth discussion; (3) resident travel options are an important factor affecting environmental issues in transportation. Carbon emission projections and analyses of emission reduction scenarios should integrate the multiple possibilities of residential preferences and policy incentives. The findings of this paper provide valuable references to the energy saving and emission reduction goals of the transportation industry, and the coordinated development of the industry and the economy.

**Keywords:** review; green development; transportation industry; environmental efficiency; low carbon



## 1. Introduction

In the current scenario, where transportation predominantly contributes to extreme pollution, energy-consuming nations are continuously striving to find ways to ensure environmental sustainability. The transportation sector, a critical link between a nation's production and consumption, is also one of the primary sources of energy consumption and carbon emissions. Since 2017, the transportation sector has emerged as the world's second-largest contributor. Despite a decrease in residents' travel needs due to the significant global public health event in 2020, energy consumption in the transportation field still accounted for about 26% of the total in 2022 [1], and carbon emissions amounted to approximately 21% (as shown in Figure 1) [2]. Projections by the IEA suggest that, by 2030, the share of $CO_2$ emissions from all sectors might rise to 50%, and, by 2050, it is expected to reach 80% [1]. Experiences from developed countries indicate that only with the transportation sector's synchronous development with the economy can the overall advantages and comprehensive benefits of the transportation system be fully realized, elevating the level of transportation development to new heights [3]. Simultaneously, the 'Green Economy Blue Book' points out that 'green development' represents a more resource-efficient, cleaner, and recoverable state of development, an active interaction between

'economy-nature-society' [4], and a state of balance between resources, environment, and economic development [5]. With the rapid growth in emerging industries, such as ride-hailing services, shared bicycles, and online freight platforms, low-carbon production and lifestyles in the transportation sector are gradually taking shape. Additionally, virtual reality technology has also been substantiated as a mitigating factor for the downturn in the tourism industry during the COVID-19 pandemic by altering consumer patterns [6]. Furthermore, it facilitates addressing the challenges of sustainable development in urban transportation [7]. This impact persists into the post-pandemic era, indicating a paradigm shift in the transportation sector. The concept of green development has permeated various dimensions of the transportation sector, redirecting its developmental trajectory. The sector is increasingly prioritizing quality and efficiency of development over mere speed and scale. A search in CNKI and Science Direct using relevant keywords reveals a significant increasing trend in research on green development in the transportation field across various countries (as shown in Figure 2), yet there is still vast room for research.

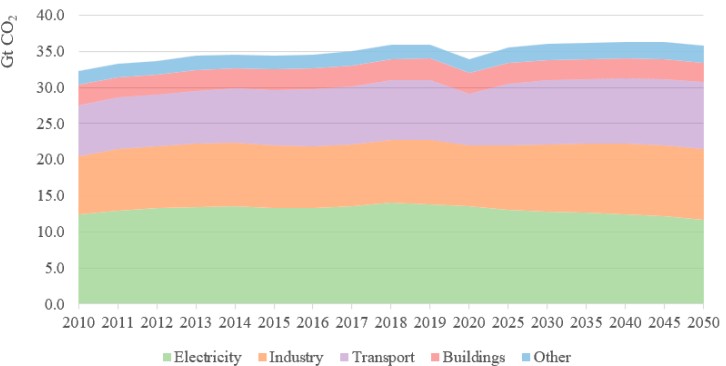

**Figure 1.** Global CO2 emissions and projections by sector.

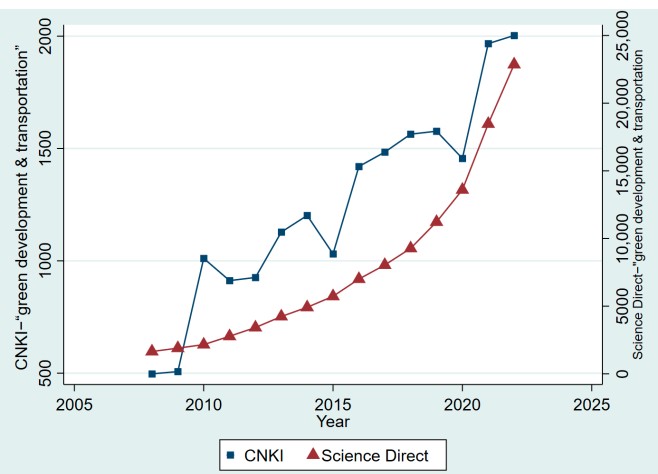

**Figure 2.** Number of papers published in journals with relevant keywords from 2008 to 2022.

The global transportation industry continues to face immense pressure to reduce emissions, particularly those of energy consumption and carbon emissions. Figure 3 shows the variations in $CO_2$ emissions from the transportation sectors of 18 typical economies [1,8–10]. Represented by the United States, France, Germany, Portugal, and the United Kingdom, the $CO_2$ emissions from the transportation sectors of many countries occupy a relatively large share of their total energy consumption carbon emissions, especially in the United States, where it accounts for 33% of national carbon emissions. Meanwhile, for many EU countries in the middle to late stages of industrialization, with relatively stable transportation service development, the transportation sector has become the only sector with continuous growth

in carbon emissions. Therefore, the development of this global high-energy-consuming industry—the transportation sector—should adapt to various urbanization needs and simultaneously reduce its negative impact on the environment to achieve ecological sustainability [11]. Global research has progressively confirmed the correlation between transportation and green development, with an increasing number of studies emphasizing the construction of more comprehensive and integrated approaches to understand the significance of achieving green development in the transportation sector.

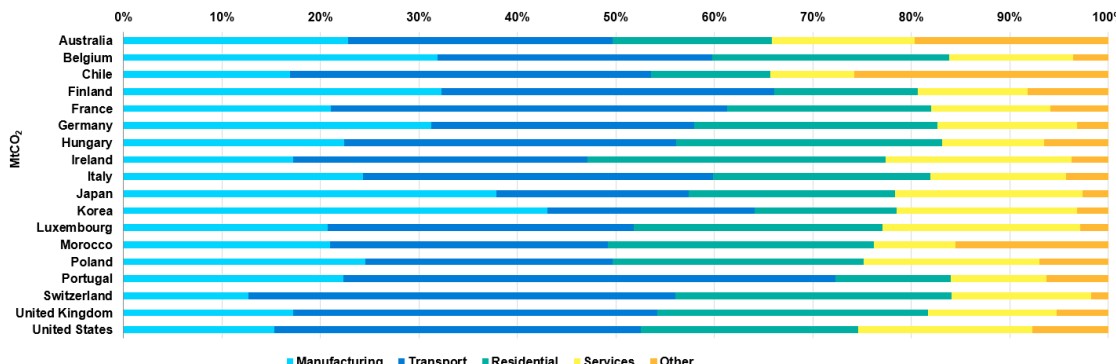

**Figure 3.** The proportion of $CO_2$ emissions from different industries in typical economies.

In addressing the issue of green development in the transportation sector, research can be categorized into three pivotal aspects: 'What', 'Why', and 'How'. Firstly, the quantification of green development indicators in the transportation sector has shifted from merely focusing on production, organizational, and service efficiencies to a more comprehensive measurement of green efficiency [12,13]. This primarily emphasizes energy consumption as an input indicator and direct carbon emissions as an undesirable output indicator. However, the carbon transfer between the transportation sector and other sectors, despite its role as a vital connector across regions and industries, is often overlooked. A contemplative approach is needed to scientifically measure the green development of transportation. Secondly, existing research has explored the key factors influencing the green development process of the transportation sector from multiple dimensions, including scale effect, structural effect, technological effect, and external policies [14,15]. However, the analysis of the correlation between technological innovation pace, market acceptance, energy consumption, and carbon emissions levels is not thorough and is influenced by the variability of short-term and long-term factors. Additionally, as the influence of resident preferences on urban transportation greening deepens, discussions of subjective factors should not be neglected. Lastly, to enhance green efficiency in the transportation sector and reinforce the importance of factors positively impacting energy conservation and emission reduction targets, the potential for carbon emission reduction in the transportation sector has garnered widespread attention among scholars globally [16]. Although various scenario simulations are increasingly refined, discussions from a full lifecycle perspective remain insufficient, and potential analyses based solely on carbon emissions lack persuasiveness.

This study aims to review the current state of research on green development in the transportation sector and to deeply explore the relationship between transportation systems and the economic–social–environmental systems. Utilizing research content, such as efficiency measurements, factor identification, and emission reduction pathway analysis, this study explores the operational and developmental potential of national or regional transportation systems. Simultaneously, it reveals the gaps and contradictions in existing research, providing direction for stakeholders seeking sustainable development pathways in the transportation industry and supporting the development decisions of governments and managers.

The remainder of this paper is organized as follows. Section 2 analyzes the efficiency indicators and quantification methods of green development in the transportation indus-

try. Section 3 discusses the factors affecting the green development of the transportation industry and the mechanisms of action. Section 4 predicts the carbon emission reduction pathways of the transportation sector under different scenarios. Finally, we draw conclusions in Section 5.

## 2. Research on the Efficiency of Green Development of the Transportation Industry

### 2.1. The Connotation and Measurement Index of Green Development Efficiency

In the realm of transportation, a global standard to gauge the sector's green development has yet to be established. However, the foundational framework typically emphasizes the measurement of efficiency values. Contemporary research has shifted focus from production, organizational, and service efficiencies towards a green efficiency paradigm. In the early stages of research, the transportation sector's economic viability was primarily assessed, with industrial output or turnover serving as the sole output metric for evaluating transportation efficiency [17,18]. Metrics, such as per unit freight turnover [19], carbon emission conversion per unit turnover [20], and carbon emission intensity [21], were employed to quantify carbon emission efficiency, thus, reflecting the production or organizational efficiency within the transportation sector under environmental impacts.

In recent years, the limitations of single-metric approaches, which assess efficiency from a solitary perspective, have become evident. These approaches fail to capture the multidimensional nature of economic green development efficiency. Consequently, there is a growing interest in harmonizing environmental and economic benefits to comprehensively reflect the actual production processes in transportation. Adopting a total factor perspective, a more intricate system for evaluating green efficiency in the transportation sector has been developed. This system incorporates capital stock, labor, and energy consumption as input metrics, with transport turnover and the added value of the transportation sector as desired output indicators, and carbon dioxide, nitrogen oxides, and sulfur dioxide as undesired output indicators [22–25]. Superseding traditional single metrics or unilaterally emphasized production function indicators, green total factor productivity and environmental efficiency indicators, which reflect the coordinated development relationship between the industry and environment, have emerged as the primary tools for quantifying the green development performance in the transportation sector. Table 1 elucidates the definitions and quantification methods of various total factor production efficiencies.

**Table 1.** Evaluation index selection and quantitative methods.

| Variables | Explanation | Quantization Method | References |
|---|---|---|---|
| Carbon emission efficiency | It is pointed out that all factors affecting carbon emission and transport turnover are factors affecting the measurement results of carbon emission efficiency. | The influence of technology and industrial structure is determined by the ratio method and the split method. | e.g., [19–21,26,27] |
| Green total factor productivity | Considering the economic benefit and environmental impact of the development of the transport industry, the input, expected output, and undesirable output ($CO_2$) indicators are used to achieve a more scientific measurement. | 1. Parameter estimation: eliminating the SFA measurement method of management inefficiency. 2. Static estimation: DEA model which can effectively avoid model errors; SBM and super-SBM models for solving factor relaxation problems. 3. Dynamic estimation: Malmquist index analysis method; Malmquist–Luenberger index analysis; MFMI analysis method. | e.g., [22–24] |
| Environmental efficiency | In order to increase the accuracy of the negative externality impact analysis in the development of the transportation industry, we expand the selection range of undesirable output indicators (e.g., $SO_2$). | | e.g., [17,25,28,29] |

### 2.2. Methods for Measuring Green Development Efficiency

2.2.1. Static Analysis Methodology

The essence of green efficiency lies in the consideration of pollution emissions and energy consumption within the efficiency of production technology. As a form of relative efficiency analysis, the measurement of green efficiency necessitates the construction of a production frontier; hence, stochastic frontier analysis (SFA) and data envelopment analysis (DEA) are commonly employed assessment methods. The SFA model, which predetermines the form of the production function, is specifically applied to the production processes of enterprises. It serves as an efficient means to eliminate the impact of managerial efficiency in measuring the efficiency of various indicators. However, it is susceptible to structural biases due to the potential misspecification of the production function [26,28,30]. The DEA model, capable of better fitting multi-output production activities that include undesired outputs and circumventing the rigid assumptions of model specification and the normal distribution of stochastic error terms inherent in SFA, finds more extensive application in efficiency evaluation from a static perspective [29]. Utilizing linear programming and convex analysis to establish the production frontier boundary, the DEA model projects different decision-making units (DMUs) onto this boundary. The relative efficiency among DMUs is evaluated based on their deviation from this frontier, where DMUs on the boundary are considered technically efficient ($efficiency = 1$), while those below it are deemed technically inefficient ($efficiency < 1$).

Moreover, a fundamental requirement of the traditional DEA model is the minimization of inputs for a corresponding maximization of outputs, making it unsuitable to incorporate environmental pollution variables. Hence, numerous methodologies have been proposed to integrate environmental pollution variables into the productivity analysis framework. For instance, multi-stage DEA models [27,31], SBM-DEA models [32–34], super-SBM models [35,36], and network DEA models [37,38] consider the impacts of random errors and relaxation of factors. By distinguishing environmental factors, random errors, and internal management variables, these extended models enhance the accuracy of measuring green development efficiency in industries, such as manufacturing and transportation. However, DEA models do not account for temporal factors and efficiency changes over time, which may lead to incompleteness in evaluation results under certain circumstances.

2.2.2. Dynamic Analysis Methodology

The DEA model is limited in its ability to reflect the trend of productivity changes and often requires a long series of empirical data to deduce the dynamic characteristics of efficiency. Thus, the Malmquist index based on directional distance functions [39–41], and its extended analytical methods, such as the meta-frontier Malmquist index analysis [42,43], have been developed to overcome the lack of dynamic perspective in efficiency evaluation inherent in the DEA model. These methods are pivotal in assessing the environmental performance of the transportation sector from a dynamic angle. Additionally, the actual production technology efficiency in the industry encompasses a wide array of variables, many of which are beyond subjective control. Research has introduced external environmental variables, such as technological advancement, environmental regulations, and asset structure, continuously refining the green development evaluation indicators for the transportation industry [44–46]. By synthesizing indicator selections and quantification methods from the domestic and international literature, the selection and quantification methods for input, desired output, and undesired output indicators have gained widespread recognition. Detailed selections and quantification methods for these indicators are presented in Table 2.

**Table 2.** Input–output index selection and quantification.

| Type | Variables | Quantization Method |
|---|---|---|
| Input | Capital stock | The investment of fixed assets in transportation, warehousing, and postal services shall be converted into 2002 comparable prices according to the price index (unit: RMB 100 million). |
| | Labor force | Number of employees in the transportation industry (unit: 10,000). |
| | Energy consumption | Consumption of electricity, heat, gasoline, natural gas, and raw coal converted into standard coal (unit: 10,000 tons of standard coal). |
| | Implied energy | Measured according to a non-competitive input–output model (unit: 10,000 tons of standard coal). |
| Expected output | Gross industrial product | Transport, storage, and postal products are converted into 2002 comparable prices using the deflator (unit: 100 million yuan). |
| | Passenger/freight turnover | Total passenger turnover (unit: 10,000 people/km). Total turnover of goods (unit: 10,000 ton-km). |
| | Social development index | According to the social dimension index system (as shown in Table 3), $P_i = \sum_{l=1}^{m} \omega_l R_{il}$, ($\omega_l$ is the weights of indicators $l$, and $R_{il}$ is the standardized value of the $l$ index in region $i$ after polar fingering). |
| Undesirable output | Carbon dioxide emission | According to the IPCC (2006), $C_t = \frac{44}{12} \times \sum_i E_t^i \times LCV_i \times CF_t^i \times O_i$. |
| | Implicit carbon | Measured by a non-competitive input–output model (unit: 1 m tons). |
| External environment | Technological progress | R&D expenditure (unit: 100 million yuan). |
| | Environmental regulation | Ratio of industrial added value to total energy consumption (unit: yuan/ton). |
| | Endowment structure | Ratio of capital stock to labor force (unit: ten thousand yuan/person). |

Note: In the carbon emission accounting formula, $C_t$ refers to the total carbon emission in year t, 44/12 is the molecular weight of carbon in carbon dioxide, $E_t^i$ represents the consumption of energy i in year t, $LCV_i$ represents the thermal equivalent of energy i, $CF_t^i$ represents the carbon emission factor of energy i, and $O_i$ represents the carbon oxidation factor of energy i.

**Table 3.** Social dimension index system.

| Target Layer | Criterion Layer | Indicator Layer |
|---|---|---|
| Social development index | Living standard | GDP per capita |
| | | Town Engel coefficient |
| | | Rural Engel coefficient |
| | Urbanization level | Disposable income of urban residents |
| | | Built-up area |
| | | Proportion of non-agricultural population in total population |
| | Transport level | Freight turnover |
| | | Passenger turnover |
| | | Traffic accident volume |
| | Scientific and educational level | Number of universities and research institutions |
| | | The proportion of expenditure on science and education institutions in the government budget |
| | | Number of patents granted |

### 2.2.3. Spatial Analysis Methodology

With the progressively widespread application of spatial econometric methods, the spatial spillover effects of carbon emissions, a crucial indicator of industrial development efficiency, have garnered extensive attention. Utilizing spatial autocorrelation analysis to construct spatial

panel models, it has been observed that technological externalities and production process dependencies in industrial development significantly influence carbon emissions due to the characteristics of neighboring regions, exhibiting notable spatial correlations [47,48]. In the case of the transportation sector, a vital inter-regional connector, its spatial dependency is even more pronounced. Numerous studies have confirmed the spatial clustering characteristics and regional disparities in carbon emissions from the transportation sector, with economically advanced regions being more affected [49–51]. Furthermore, to explore the structural characteristics of total carbon emissions after integrating spatial correlation effects, as well as to delve deeper into the spatial patterns and evolution of carbon emissions across different industries and sectors [52], social network analysis has begun to be applied in studying the spatial correlation networks of carbon emissions. Many scholars have employed this methodology to construct networks concerning specific areas, such as aviation [53,54], urban public transportation [55,56], industry-wide mobility [57], and regional carbon emission networks [58,59], to simulate and analyze the spatial evolution of carbon emissions.

## 3. Research on the Mechanisms of Green Development in the Transportation Industry

### 3.1. Analysis of Factors Affecting the Green Development of the Transportation Industry

3.1.1. Explanation of Influencing Factors

Identifying key factors influencing green efficiency in the transportation sector is critical for formulating development strategies that yield both environmental and economic benefits. Although the focus varies among countries, commonly, factors influencing industrial green development are screened from three aspects: industrial scale, structure, and technological innovation [58–63]. Taking China's 30 regions as an example, Figure 4, influenced by regional heterogeneity, reveals significant differences in undesired outputs, specifically carbon emissions, across different regions. Simultaneously, using transportation data from 2005 to 2020, the impact proportion of each factor on the total effect was analyzed. Figure 5 indicates that the scale effect plays the most pronounced role in promoting carbon emissions, while structural and technological effects tend to suppress emissions, with the impact of industrial structure being more significant. Explanations for these various influencing factors are provided in the following text.

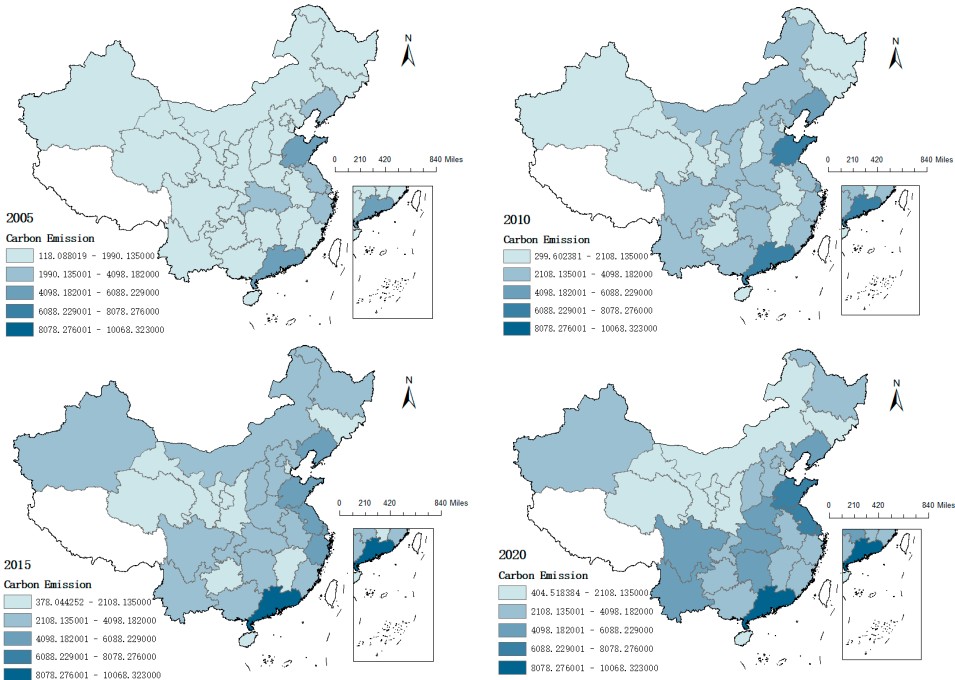

**Figure 4.** Partial annual CO$_2$ emission of the transportation industry in 31 regions of China.

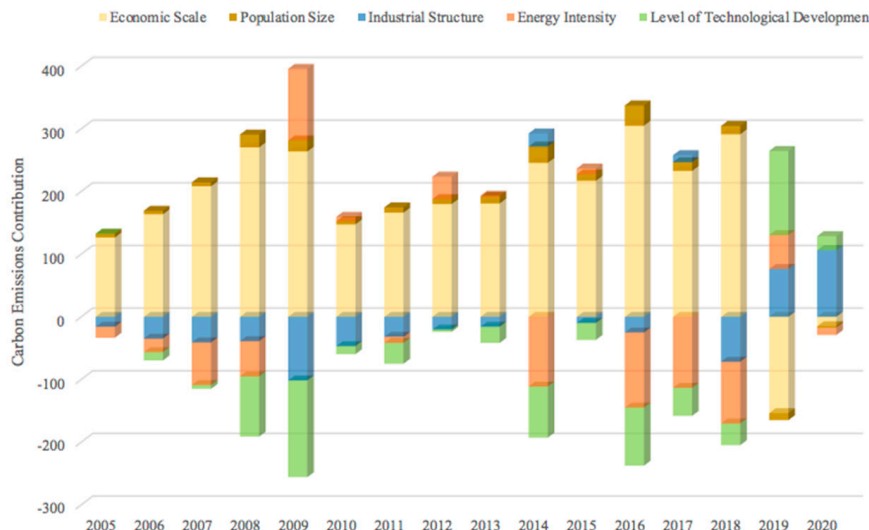

**Figure 5.** Contribution of carbon emissions from different factors (2005–2020).

For most developing countries and some developed nations, scale effects trigger expansions in population size, per capita GDP, transportation infrastructure scale, and economic scale. Consequently, energy consumption and carbon emissions increase with expanding scale, hindering the green development of the sector to some extent [64–71]. Specifically, in the past 30 years, the transportation sector in developed countries has generally had higher total and per capita carbon emissions compared to developing countries. Hence, nations can be categorized into 'advanced economies', 'emerging markets and developing countries', and 'others'. Figure 6 displays typical indicators affecting the scale effect of these three categories and their shares of carbon emissions, revealing a direct correlation between larger GDP and population size and higher carbon emissions [1]. However, due to the difficulty in rapidly adjusting indicators, like economic development level and population size, the restraining effect of scale expansion on energy conservation and emission reduction in the transportation sector remains significant.

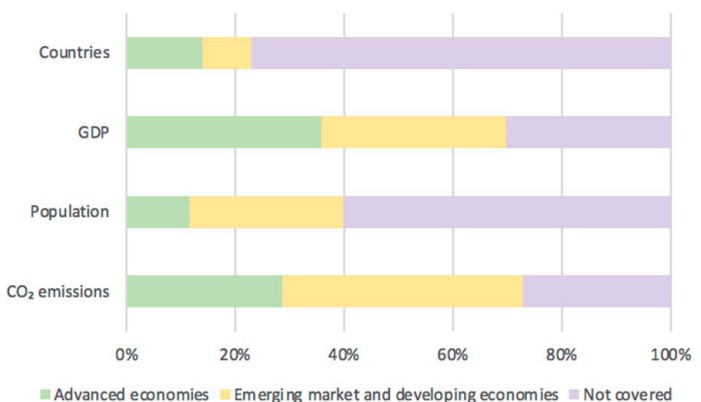

**Figure 6.** Proportion of relevant indicators in different types of countries.

Moreover, optimizing transportation intensity and service structures related to industrial development also serves as an effective means to curb energy consumption and carbon emissions [72–75]. Road transportation, which accounts for approximately three-quarters of the total carbon emissions from the transportation sector, is key to achieving the sector's zero-carbon goal [76,77]. Figure 7 reveals that road transport, including passenger and freight, contributed 77% to the global transportation sector's carbon emissions in 2021, with the remainder emanating from the aviation, maritime, and rail sectors. According to data from the International Energy Agency shown in Figure 8, road transportation indeed

generates significantly higher CO2 emissions compared to other modes of transport [1]. Therefore, identifying the optimal transport mix and clarifying the optimal transport modal split will be key to curbing energy consumption and emissions.

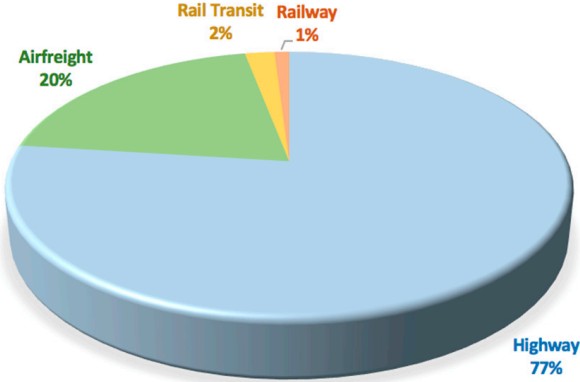

**Figure 7.** Share of carbon emission from four common modes of transportation worldwide.

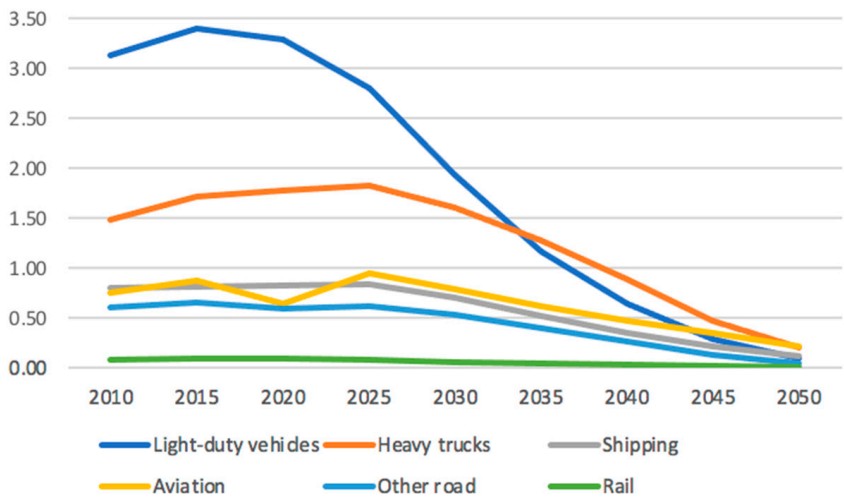

**Figure 8.** Global $CO_2$ transport emission by mode.

In terms of industrial technological innovation, increased R&D investment, reallocation of intangible innovative factors (such as human and knowledge capital), and promotion of new energy technologies are crucial tools for enhancing energy use efficiency [78–80]. However, the presence of the energy rebound effect obscures the relationship between technological progress-induced energy efficiency changes and overall carbon emissions and green development in the industry. Hence, accurately measuring the energy rebound effect is vital for enhancing the effectiveness of energy consumption reduction in the transportation sector. In recent years, rebound effect research has received significant attention in high-energy-consuming industries, particularly transportation, to avoid the insufficient motivation and contribution of technological innovations and the ineffectiveness of energy-saving and emission-reduction policies [81–84].

Finally, external regulatory policies, such as the implementation of new environmental laws and the introduction of carbon emission trading markets, are also crucial in aiding the transportation sector in achieving coordinated economic and environmental development. In the short term, environmental regulation increases costs for pollution-intensive industries, squeezing the space for productive investment and technological innovation. High-pollution enterprises are forced to relocate or exit the market, leading to a 'pollution haven' effect due to lower costs, ultimately impacting industrial development outcomes. Meanwhile, existing technological levels and production demands

remain unchanged [61,85]. In the long run, based on the 'Porter Hypothesis', environmental regulations will compel industries to engage in technological innovation through adjustment and improvement of production strategies. By incentivizing investment in technological innovation, directing more resources into developing green technologies and production equipment, and forming innovation compensation mechanisms to offset or even surpass the 'compliance costs' brought by environmental regulations, a win–win situation for industrial development, energy conservation, and emission reduction can be achieved [86]. Therefore, it is important to note that external policies may have a lag effect on the impact of green development in the transportation sector. Table 4 systematically summarizes the factors affecting green development in the transportation industry and the related literature sources.

**Table 4.** Research on influencing factors of green development of transport industry.

| Type | Variables | Relevant Research |
|---|---|---|
| Scale effect | Population size | e.g., [58,63,64,71] |
| | GDP per capita | e.g., [65,74,87,88] |
| | Passenger/freight turnover | e.g., [68,89,90] |
| Structural effect | Private car ownership | e.g., [60,61,71,91] |
| | Sharing rate of public transport | e.g., [84,92–94] |
| | Energy consumption structure | e.g., [66,73,77,95] |
| Technical effect | Energy intensity | e.g., [78,80,96] |
| | Emission factor | e.g., [75–77,79] |
| | Energy rebound effect | e.g., [81–84] |
| External constraint | Environmental governance/protection policy | e.g., [85,86,97,98] |

3.1.2. Factor Decomposition Methodology

Identifying and quantifying the various factors influencing the green development of the transportation sector is crucial for prioritizing development strategies. Factor decomposition methods have been widely acknowledged by scholars globally. Initially, the IPAT model demonstrated that population, affluence, and technology are the three primary factors affecting environmental conditions and industrial development [99]. The stochastic environmental impact assessment model (STIRPAT) strictly constrains the measurement of the degree of influence of various factors, such as population, level of economic development, energy efficiency, transportation structure, and level of urbanization, on the carbon emissions of the transportation industry in different environmental contexts [92,100–104]. It overcomes the limitation of the IPAT model, which relies on the equal proportional influence of variables, because it is easier to obtain quantitative relationships between variables. However, this method still suffers from a lack of flexibility. Additionally, the structural decomposition analysis (SDA) method based on the input–output model [89] and the production theoretical decomposition analysis (PDA) method [91] are frequently used to discuss the impacts of industrial scale, structure, and technology effects on carbon emissions. Nevertheless, these methods fall short in precisely measuring structural factors, like economic structural effects and energy consumption, potentially leading to conclusions contrary to actual situations. In such cases, the index decomposition analysis (IDA) method based on time series data is widely used. Furthermore, the logarithmic mean divisia index (LMDI) decomposition method, known for its simple decomposition form, independence from the decomposition factors, and zero residual in decomposition results, effectively addresses the zero-value problem and has become one of the preferred choices in resource and environmental research [87,93,97]. The frequency of three types of factor decomposition models used in green development research in the transportation sector, as found on China National Knowledge Infrastructure (CNKI) and Science Direct, is illustrated in Figure 9.

However, in using factor decomposition methods to identify key factors and measure their impact, the autocorrelation among factors and regional spatial autocorrelation are often overlooked. Future research should focus on enhancing the application of spatial econometric models and constructing hybrid models to improve analytical accuracy. This will assist different regions in the transportation industry to better formulate energy-saving and emission-reduction policies.

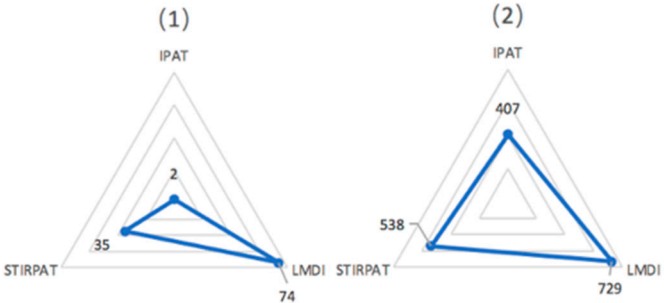

**Figure 9.** Frequency of various research methods in CNKI (1) and Science Direct (2).

### 3.2. Analysis of the Impact Pathways for Green Development in the Transportation Industry

In the context of exploring the mechanisms influencing green development in the transportation sector, Figure 10 presents a delineated pathway of impact, offering a reference for future developmental trajectories across different regions. Specifically, the scale effect encompasses economic, population, and industrial dimensions. Rapid economic growth and urban population increases lead to surging transportation demand and an expanding transportation market, consequently elevating pollutant emissions and manifesting adverse environmental impacts [78,105]. However, the expansion in investment scale in the transportation sector also provides ample financial support for urban infrastructure construction. Developments in charging and refueling facilities and projects, like 'oil-to-electricity and oil-to-gas' conversions for loading machinery and transportation equipment [88,106], reduce cross-regional factor mobility costs, enhancing industrial economic output efficiency and clean energy utilization of machinery. Effective allocation of transportation infrastructure may also exhibit a positive influence on energy conservation and emission reduction [107,108].

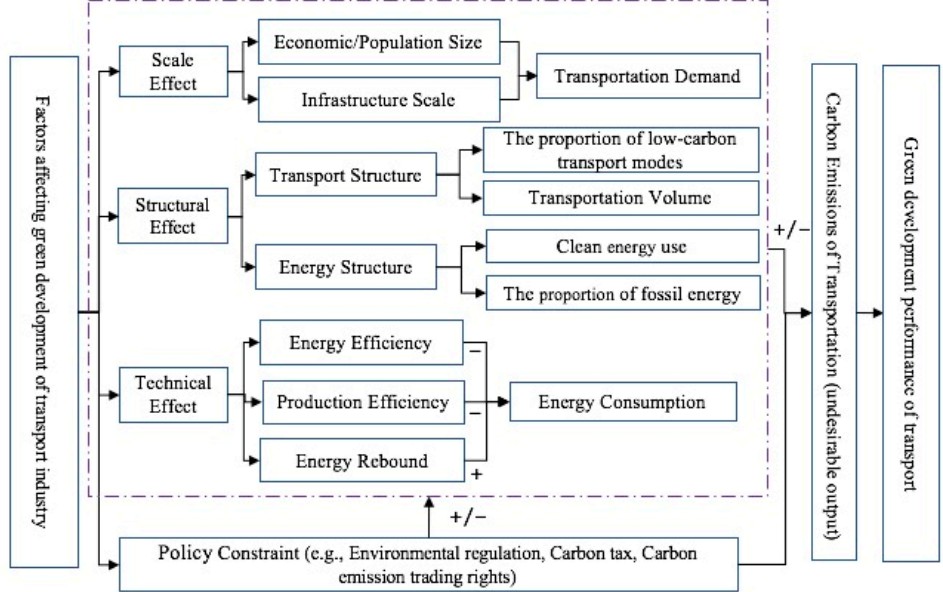

**Figure 10.** Impact mechanisms of factors affecting green development of transport industry.

Structural effects are primarily manifested in the impacts of transportation and energy structures. As a critical input indicator in assessing green efficiency in the transportation sector, the proportion of fossil fuels in energy consumption alters the sector's energy structure [102]. An increased proportion of high-carbon-emitting sources, like coal, gasoline, and diesel, diminishes carbon emission efficiency [98,109]. Thus, the vigorous promotion of green travel concepts, along with the proliferation of clean energy and public transportation modes, has increased the proportion of low-carbon travel, subsequently enhancing the sector's green efficiency [110,111]. Moreover, the development of the transportation sector promotes effective resource redistribution, reducing the proportion of 'high pollution, high emission' industries in primary and secondary sectors. However, improvements in the service quality of the transportation sector may transfer high energy consumption and pollution characteristics to the sector, potentially exerting adverse impacts on its green development [108,112]. Therefore, as the proportion of service industries, like transportation, increases, the expansion of manufacturing economies indirectly drives up demand in the transportation sector. Structural optimization also faces the risk of stimulating industrial scale, thereby promoting energy consumption and carbon emissions.

Furthermore, research indicates that technological advancements in the transportation sector are key to reducing or even curbing pollutant emissions and lowering energy consumption [113–115]. Relying solely on repetitive labor without more profound technological innovations keeps the high productivity of resources in flux. Intensifying R&D investments and fostering technological innovations in the transportation sector, along with accelerating the effective transformation of technological achievements, will enhance resource utilization efficiency and drive future green growth in the sector [116,117]. Firstly, in recent years, transportation intelligence, encompassing vehicle electrification, networked roads, and shared routes, has directly impacted industry production and energy utilization efficiencies [118,119]. Simultaneously, enhanced technological innovation capabilities indirectly stimulate industrial carbon emissions and efficiency through economic growth and industrial structure optimization [95,120]. Also, as a common indicator of technological level, energy intensity is a crucial factor restricting the growth of carbon dioxide emissions. The introduction of clean production technologies and improvements in energy utilization techniques can indirectly enhance energy utilization efficiency through scientific planning of energy input quantities and effective energy input ratios, thereby further advancing the low-carbon industrial development process [121,122]. Additionally, the ongoing technological revolution led by the digital economy accelerates the deep integration of transportation and industry, with digital transportation offering effective pathways to promote green industrial transformation [94,123]. However, as industrial digital transformation is still in its nascent stages, related research is not extensive and merits further discussion.

On the other hand, while technological progress can promote the synergistic development of carbon emission reduction and economic growth, improving energy efficiency [124], it may also lead to the 'Jevons Paradox: Energy Rebound Effect' [96,125]. The rebound effect implies that the anticipated energy-saving effects from improved energy efficiency are offset by increased energy demands, diminishing the effectiveness of environmental policies [126,127]. Studies confirm the impact mechanism of the energy rebound effect on industrial development and the variability of different paths of energy rebound effects through internalizing energy efficiency and simulating the impact of different types of energy efficiency improvements on energy consumption [128–131]. High-energy-consuming sectors, like the steel [132], construction [133], and transportation [126] industries, are significantly affected by the energy rebound effect. Moreover, in some countries, like Denmark [134], Norway [135], the United Kingdom [136], and China [129], the energy rebound effect in the transportation sector typically encompasses long-term and short-term effects. In quantifying the energy rebound effect, academia attempts to define the energy-saving effect of improved energy efficiency as the elasticity of energy consumption to energy efficiency, with the value of the energy rebound effect being the elasticity of energy consumption to energy efficiency plus one. Due to the complexity of measuring

energy efficiency, the elasticity of energy consumption to energy prices and the elasticity of energy services to energy prices have also been used for quantitative analysis of the energy rebound effect, rather than the elasticity of energy consumption to energy efficiency [81,83,137]. However, most studies have not emphasized the impact of the endogeneity of technological progress on results. Indeed, introducing biased technological progress parameters into the production function will enhance the scientific validity and effectiveness of measuring the energy rebound effect.

Finally, the importance of external constraints cannot be overlooked. Although research on the environmental regulation and green industrial development has not yet formed a unified framework, studies recognize that environmental regulation exhibits both inhibitory effects on green development efficiency [90] and positive influences on ecological environment and high-quality industrial development [138]. As it is affected by policy lag effects, the impact of environmental regulation on the green development of industries, like manufacturing and high-pollution sectors, may shift from hindrance to promotion [139]. Among them, the impacts of command-and-control, market incentive, and public participation forms of environmental regulation on industrial green growth vary. The direct impacts of market incentive and voluntary agreement types of environmental regulation on industrial green total factor productivity are, respectively, inverted U-shaped and U-shaped relationships. In contrast, command-and-control environmental regulation has not directly affected green total factor productivity [140,141]. Regionally, high-level areas primarily adopt the method of selling emission rights while raising emission fee standards for industrial green development, whereas low-level areas emphasize accelerating the transition from command-and-control to market-incentive environmental regulation to enhance the effect of external policy constraints [142].

## 4. Research on the Prospect of Green Development of the Transportation Industry

### 4.1. Research on the Impact of External Policy Constraints

With rapid economic development and continual advancements in technology, the transportation sector is urgently required to decarbonize to fulfill the net-zero commitments within the economic sphere. According to the International Energy Agency (IEA), global urban passenger transport carbon emissions are projected to grow at an average annual rate of 1.7%, with developing and transitional economies expected to reach 3.4% and 2.2%, respectively, by 2030 [1]. Figure 11 displays the carbon emission projections for the transportation sector across different types of countries, indicating that both technologically advanced and developing countries are indeed facing significant carbon reduction challenges in the long term [143].

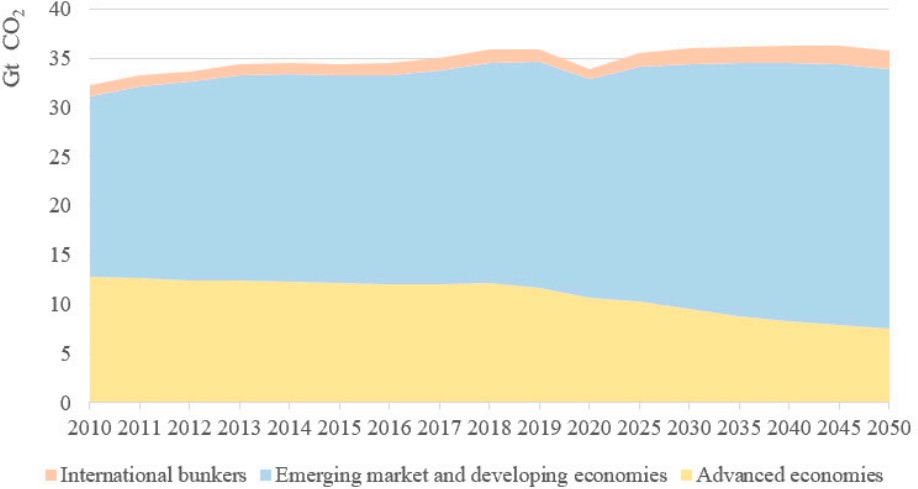

**Figure 11.** Energy-related and industrial process $CO_2$ emission by region.

Given the variations in resource, environmental, and economic development charac­teristics of cities, countries worldwide may pursue diverse development modes, such as the 'low-carbon society' comprehensive goal model, the internal pull mode of low-carbon industries, the 'point-to-area' demonstration development model, and the 'low-carbon supporting industry' development model [144,145]. Hence, to achieve more accurate and scientific pollution emission predictions, research initially emphasizes the importance of external policies and management measures in countries and regions. Considering the high energy consumption and high emissions characteristic of the transportation sector, related studies often explore decarbonization from four perspectives—energy decarbonization, production decarbonization, consumption decarbonization, and emission decarbonization (as illustrated in Figure 12)—discussing how to coordinate industrial development with environmental protection to the maximum extent through external policy constraints.

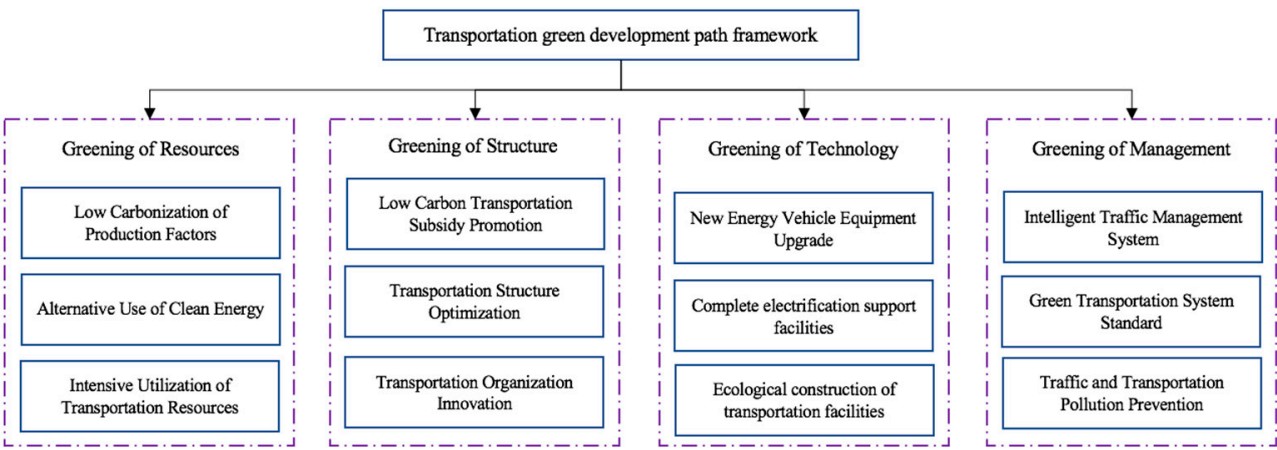

**Figure 12.** Framework of the transportation green development path.

Studies have found that formulating green, low-carbon transportation plans, while taking into account aspects, such as transportation subsidies [146,147], energy consumption control [148,149], and sustainability perspectives [150], plays a crucial role in promoting new energy vehicles, optimizing transportation structures, and advocating for green travel modes. Research based on behavioral economic principles (such as loss aversion effects) analyzed the dual impacts of public transport and taxi fare adjustments [151], environmen­tal taxes, carbon taxes [152], and consumption tax collection based on vehicle types and emission standards [153] on travel. These studies avoided the dual inflection point effects, weak emission reduction effects, and rebound effects of early single congestion charging policies on pollutant reduction. By constructing clear vehicle travel decision algorithms, we explored the differences in emission reduction among different collaborative strategies. Moreover, green transportation schemes led by public transportation, supported by vehicle electrification, and complemented by walking, cycling, shared transportation, and other slow traffic are being implemented in various countries. Therefore, reducing urban trans­portation energy consumption relies not only on ordinary vehicle traffic restrictions and lottery policies [154], but also on scientifically reducing industrial energy consumption, supplemented by the implementation of various subsidy policies, and urban shared trans­portation schemes are increasingly playing a positive role [155,156]. Finally, governments are continuously promoting transportation energy-saving and emission-reduction technolo­gies [157], as well as low-carbon travel concepts, such as new energy vehicles [158,159]. By curbing rapid and significant increases in transportation energy consumption and carbon dioxide emissions, these efforts aim to enhance the environmental benefits of the transporta­tion industry and promote its coordinated development with environmental protection. Some typical countries, such as China, the United States, Germany, and the United King­dom, still persist in enhancing their economic growth by utilizing non-renewable energy

resources based on fossil fuels. The pace at which these economies are moving towards greener or cleaner energy production is far greater than that of other global energy users. Therefore, this study focuses on analyzing the impact of these countries' policies on the green development of the transportation sector. Table 5 reveals specific policy measures taken by regions with a high share of transportation sector emissions, led by the United States, to achieve green development goals in transportation.

**Table 5.** Typical national measures to promote green development in the transportation industry.

| Country \ Objectives | Formulate a Strategic Plan for Green and Low-Carbon Transportation | Promote New Energy and Clean Energy Vehicles | Optimize Transportation Structure | Advocate Green Travel |
|---|---|---|---|---|
| USA | "The Transportation Security Act", "Clean Air Act", "Multimodal Transportation Act", etc. | Introducing "The Energy Policy Act" provided a $3400 tax credit for new hybrid light-duty vehicles | Establishing multimodal transportation development policies | Charging high parking fees |
| EU | "The Directive on the Establishment of General Guidelines for Multimodal Transport between EU Member States", "The Sustainable and Intelligent Transport Strategy", etc. | Making full use of digital technologies to make travel and mobility smarter, more efficient, and more environmentally friendly | Launch of "the Marco Polo program", with financial subsidies and tax breaks | Increase in fuel tax rates |
| Germany | "The National Bicycle Transportation Plan", "The Urban Transportation Finance Act", "The Structural Strengthening Act", etc. | Raising vehicle emission standards; allowing new energy vehicles to use bus lanes | Implementing an innovative combination of classic overhead lines and train-driven energy alternatives; redefining electrification rates | Establishing restrictive policies for car use |
| UK | "Decarbonizing Transport, A Better, Greener Britain", "The Green Industrial Revolution", etc. | Increasing the cost of car ownership and use; improving electric vehicle infrastructure; charging for excess emissions | Greater investment in green transport, such as bicycle riding and eco-buses | Designating congestion charging areas and levy vehicle taxes |
| China | "The 14th Five-Year Plan for Green Transportation", "The Green Transportation Standard System", etc. | Adopting a time regression mechanism and increase government subsidies for new energy vehicles, such as subsidies and free parking | Improving the operational efficiency and service level of public transportation, such as buses, subway, and light rail | Adopting some traffic restriction policies; charging congestion fees; environmental protection publicity and education training |

### 4.2. Prediction Methodology

For the design and implementation of green development in urban transportation, understanding and analyzing the interactions among a range of dynamic factors that shape transportation patterns, behaviors, and impacts is crucial. System dynamics models (SD) facilitate in-depth studies of the complex system composed of transportation, socio-economic, energy, and environmental components and simulate predictions of the effectiveness of

green development strategies implemented in the transportation sector [160–162]. SD models are composed of stocks, flows, and auxiliary variables and are used to analyze complex dynamic feedback systems. They graphically express the interactions among various factors in the transportation carbon emission system through stock-flow diagrams [163–165]. Owing to their proficiency in handling nonlinear, high-order, multivariate, multi-feedback, and cyclical system issues, system dynamics methods are widely applied in studies on energy conservation and carbon emissions, including in socio-economics, primary, secondary, and tertiary industries, housing, transportation, waste management, and electricity [166,167]. The transportation industry carbon emission system is a complex system with dynamic changes, typically divided into economic, energy, and environmental subsystems, or road, rail, and waterway subsystems. Consolidating past research, Figures 13 and 14, respectively, represent the causal loop diagram and system flow diagram of the complex transportation–emission system, revealing the paths of energy consumption and pollution emissions in the transportation sector. Variables in the system can be categorized based on their attributes and significance into auxiliary, rate, and level variables. Auxiliary variables represent relationships between main variables, such as energy consumption in the transportation industry and environmental regulatory policies. Rate variables refer to the inputs and outputs of level variables, indicating changes in level variables over time, such as the share of the tertiary industry and the GDP growth rate. On the other hand, state variables represent significant stock variables, like GDP, average annual population, and carbon emissions. Ultimately, governments can simulate energy-saving and emission-reduction pathways in the transportation sector and predict pollution emission outcomes under current scenarios by combining local environmental protection and green industrial development policies, utilizing parameter setting results [168–170].

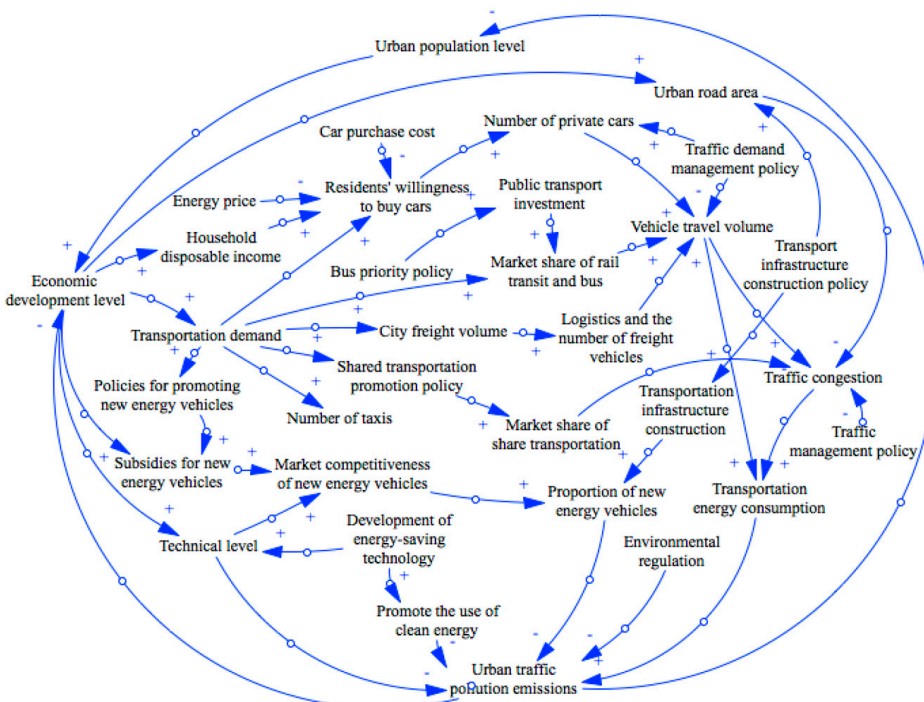

**Figure 13.** Causal loop diagram of the transportation carbon emission system.

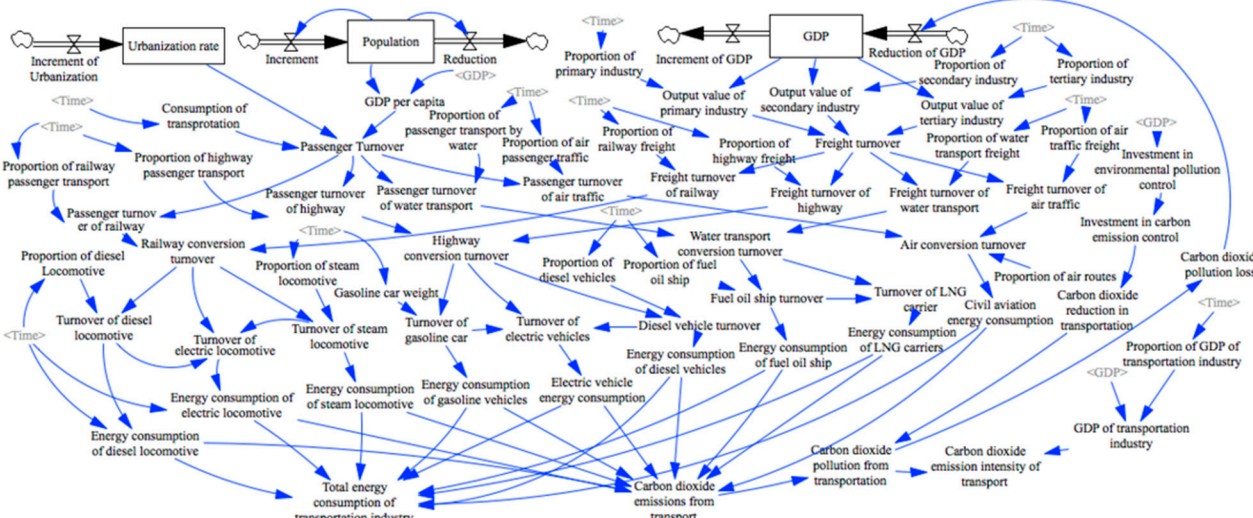

**Figure 14.** Stock-flow diagram of the transportation carbon emission system.

The long-range energy alternatives planning system (LEAP) models energy supply, utilization, and conversion technologies by modeling specific technical and economic parameters. Given its capability to quantitatively analyze the dynamic impacts of energy policies on the environment and forecast long-term energy demands and pollution emissions, LEAP is widely applied across various scales, from national to regional and industrial levels, particularly in certain sectors, like industry [171,172], transportation [173,174], and commerce [175]. Focusing on energy and carbon emissions, research primarily concentrates on future emission reduction potentials and pathways towards low-carbon transformation, emphasizing the significant impact of enhancing energy efficiency and changing energy structures on regional green development. Although the LEAP model effectively simulates energy consumption situations and their environmental impacts, it falls short of fully capturing the potential socio-economic benefits. The notable impacts of cross-industry energy transfer and emerging energy transformations on traditional development models, especially in developing countries, require further investigation [176]. Based on the scale, structural, technological effects, and external constraint policies impacting the green development of the transportation sector, this study, drawing on a review of the relevant literature, proposes three scenarios (baseline, low-carbon, and high-carbon), as shown in Table 6. The low-carbon scenario includes sub-scenarios, like structural optimization, technological progress, environmental policy constraints, and coordinated economic–environmental development. Given that industrial investment scale, energy intensity, energy structure, intensity of technological investments, and environmental regulation indicators play varying roles in different scenarios, predictions of carbon emissions and policy implementation outcomes inevitably differ. Building on the research by Yang et al. (2021) [177] and IEA data, Figure 15 illustrates the carbon emission outcomes predicted using the LEAP model under different scenarios. In the low-carbon scenario for the transportation sector, the growth and peak of carbon emissions are consistently lower than other scenarios, more in line with energy conservation and emission reduction policy requirements. Therefore, for high-pollution industries, like transportation, optimizing industrial structures, enhancing technological innovation levels, and formulating reasonable environmental regulatory policies are key to promoting coordinated development between industry and the environment and achieving a green development path [178–180].

**Table 6.** Scenario setting for the development of transportation industry.

| Scenario Hypothesis | | Scale Effect | Structural Effect | | Technical Effect | Policy Constraint |
|---|---|---|---|---|---|---|
| | | Proportion of Industrial Investment Scale | Energy Intensity | Energy Structure | Science and Technology Investment Intensity | Environmental Regulation |
| | Baseline | Medium | Medium | Medium | Medium | Medium |
| Low carbon | Structural optimization | High | Medium | Medium | Medium | Medium |
| | Technological progress | Medium | Medium | Medium | High | Medium |
| | Environmental policy constraint | Medium | Low | Low | Medium | High |
| | Coordinated development of economy and environment | High | Low | Low | High | High |
| | High carbon | High | High | High | High | High |

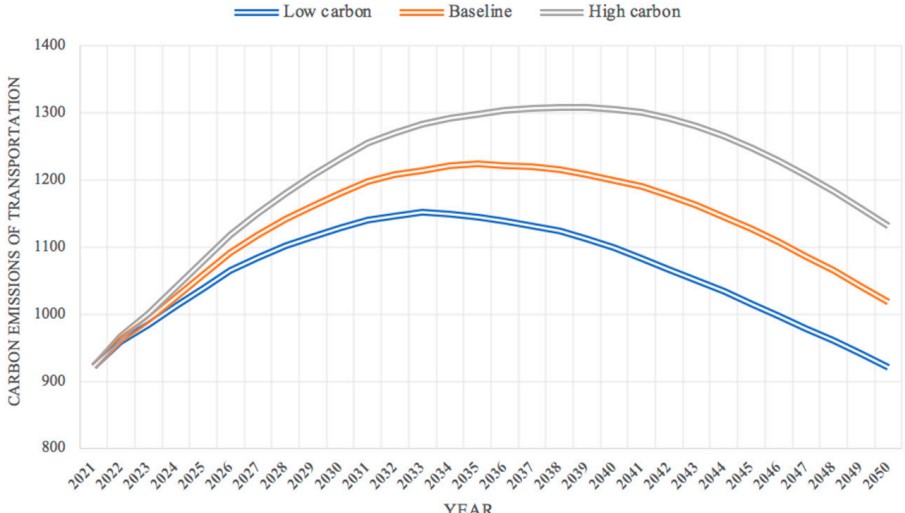

**Figure 15.** Forecast of traffic carbon emission trends under different scenarios.

## 5. Conclusions

Excessive emissions and excessive energy consumption have heightened global awareness of the necessity to strengthen green development in multiple countries. Green development in the transportation sector aims to reduce operational energy consumption in industrial development and to control associated carbon emissions. Within the burgeoning actions for green industrial development, the transportation sector has become a primary target for improvement. Although some countries, led by China, still have a significant impact on the environment through their transportation sector, this sector has been able to identify crucial factors influencing its green development through the use of reasonable environmental performance measurement methods. Utilizing scenario analysis and other simulation methods, it explores future paths for energy conservation and emission reduction.

With the development of the transportation sector, the environmental impacts of issues, such as the carbon emissions and energy consumption of vehicles, are becoming increasingly evident. Researchers, governments, and stakeholders have shown a growing interest in understanding the mechanisms that influence the greening of the transportation sector. This study reviews the current state of research on green development in the transportation sector from three perspectives: development performance assessment,

analysis of influencing factors, and exploration of development pathways. This systematic review provides a comprehensive framework for understanding sustainability issues in the transportation sector, offering robust support for government and managerial decision-making. Simultaneously, by revealing the gaps and contradictions in existing research, this paper provides clear guidance for the direction and focus of subsequent studies, further deepening and refining knowledge in this field. Specifically, the conclusions of this study include the following aspects.

Firstly, this study presents the widely recognized input–output analysis framework in the transportation sector, focusing on the estimation methods for unintended outputs, like carbon emissions, and energy–environmental efficiency. However, there is a lack of internationally accepted methodologies for calculating transportation carbon emissions specific to different urban road design plans, modes of transport, and transportation vehicles. Environmental benefit assessments in the transportation industry predominantly measure direct energy consumption and carbon emissions during the developmental process, overlooking indirect energy consumption and pollutant emissions that could arise from inter-industrial interactions within national economic development. For example, these might include carbon emissions embedded in intermediate products and services consumed by the industry and inter-industry carbon transfer. Hence, the total carbon emissions and efficiency measurement results are not sufficiently accurate, which directly impacts the setting of carbon intensity targets. Moreover, in the evaluation of green development effects in the transportation sector, the literature based on static analyses at the national level or dynamic analyses combining provincial and municipal geographical locations is not entirely reliable. Considering spatial correlations, the geographical spatial dependency characteristics of green development should be incorporated into the research framework.

Secondly, in the realm of researching factors influencing the green development of the transportation sector, a systematic compilation has been conducted encompassing the aspects of influencing factors, the extent of that influence, and the pathways of impact. However, due to the influence of the energy rebound effect and environmental regulations, the development directions of industrial energy consumption, pollutant emissions, and environmental benefits are not fixed. With the increasing prominence of technological levels as the core driver of the industry's green transformation and the growing demand of residents for the intelligence and greening of the transportation sector, research should intensify the analysis of the impact of technological innovation levels and resident preferences on advancing the greening process of the transportation sector in the future.

Thirdly, as the internal structure of the transportation energy consumption and carbon emission systems is dissected, it is necessary to explore the degree of influence of various factors and to simulate the improvement path of energy saving and emission reduction. The optimization of industrial structure, technological innovation, and economic scale have become more significant in the green development of the transportation industry. However, the most efficient means of reducing carbon emissions remains uncertain. Low-carbon travel options for residents can rapidly reduce carbon dioxide emissions at extremely low or even negative costs. Whether accelerating the development of intelligent transportation through source planning or leveraging technological advancements to empower the transformation of transportation energy structures, carbon reduction activities in the transportation sector largely depend on residents' decisions for green travel. Coordinating consumer preferences with new technology applications, implementing green development concepts and requirements in the transportation industry, enhancing the quality and efficiency of transportation development, and optimizing the development layout of the transportation industry should be further explored.

Finally, by establishing scenarios that align with the developmental needs of the transportation sector, past research has achieved simulation analyses of the internal structure of the transportation energy consumption and emission system under the influence of transportation management policies and environmental constraints. Low-carbon scenarios, which incorporate conditions, such as structural optimization, technological innovation,

and policy constraints, have indeed resulted in lower carbon emissions and the earlier achievement of a carbon peak. However, from a life-cycle perspective, the lifetime mileage of each transportation system is seldom incorporated into scenario analysis studies, leaving a knowledge gap in scenario simulations for sustainable development plans for different vehicle types. Current scenario settings based on baseline, high-carbon, and low-carbon are still somewhat rudimentary. More specific parameter settings should be considered to formulate more scientifically-based emission reduction pathway enhancement schemes for the transportation sector based on effective policy combinations. Additionally, scenario analyses mainly rely on predictions of energy consumption or carbon emissions. Comparing environmental performance or green total factor productivity across different scenario models could provide more valuable information for the future pursuit of the intelligent and sustainable development of the transportation sector.

**Author Contributions:** Conceptualization, Y.M. and X.L.; methodology, Y.M.; formal analysis, Y.M. and X.L.; investigation, Y.M. and X.L.; data curation, Y.M.; writing—original draft preparation, Y.M.; writing—review and editing, X.L.; visualization, Y.M.; supervision, X.L.; project administration, X.L.; funding acquisition, X.L. All authors have read and agreed to the published version of the manuscript.

**Funding:** This research was funded by the National Natural Science Foundation of China (51778047).

**Institutional Review Board Statement:** Not applicable.

**Informed Consent Statement:** Informed consent was obtained from all subjects involved in the study.

**Data Availability Statement:** The data presented in this study are available on request from the corresponding author.

**Acknowledgments:** We thank the editor and all anonymous referees for their comments. Additionally, we are grateful to our research group members for their valuable comments. Despite assistance from many sources, any remaining errors in the paper are the responsibility of the authors.

**Conflicts of Interest:** The authors declare no conflict of interest. The funders had no role in the design of the study; in the collection, analyses, or interpretation of data; in the writing of the manuscript, or in the decision to publish the results.

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
