# Peer review of "A Review of Research on the Impact Mechanisms of Green Development in the Transportation Industry"

_sustainability, doi:10.3390/su152316531_

Round 1
Reviewer 1 Report
Comments and Suggestions for Authors
As a reviewer of the first view, I have several major concerns. So, the authors can improve or justify strongly in order to be published in high-quality journals such as Sustainability.
- The paper is too long, you may reduce the length by providing tables.
- I can not see the methodology and strategy to search and find papers related to the topic.
- The contribution of the paper is not clear crystal.
-The conclusion and findings are not accurate enough to be suggested to other researchers.
- Some of the figures are not clear to the reader. please revise them.
Comments on the Quality of English Language
The English language is good, however, minor checking is a must.
Author Response
Thank you for dedicating your valuable time to review our survey paper and for providing insightful feedback. We greatly value your comments and have thoroughly revised our manuscript in accordance with your suggestions.
For each point of feedback you raised, we have carefully considered and made corresponding revisions in the manuscript. These changes have been clearly marked in the resubmitted document with track changes for your ease of review.
In revising our paper, we have strived to ensure all information and viewpoints are up-to-date and accurate, while also improving the logical flow and readability of the text.
We look forward to your further feedback and hope that our revisions meet your expectations. Once again, thank you for your contribution and guidance towards improving our work.
Point-by-point Response to Comments and Suggestions for Authors
Comments 1:
The paper is too long; you may reduce the length by providing tables.
Response 1:
Agree. In response to your comment, we have carefully reviewed the entire manuscript to identify areas where we can reduce length without compromising the quality and integrity of the research. This review involved:
Eliminating redundant or less critical information.
Streamlining the text to improve clarity and brevity.
Revising certain sections to convey the same ideas with fewer words.
The revised manuscript, which we believe aligns better with the journal’s standards for conciseness, has been resubmitted for your review. We hope that these modifications satisfactorily address your concerns regarding the length of the paper.
Comments 2:
I cannot see the methodology and strategy to search and find papers related to the topic
Response 2:
Thank you for your comment, we agree with your opinion. Since this paper hopes to provide a systematic review of how the transportation industry can be greened in the future, this review consists of three parts. It is hoped to express an overview of the broad concept of green development in the transportation industry by assessing the green efficiency of transportation components, factors affecting green efficiency including carbon emissions, and analyzing the path of energy saving and emission reduction in the future. Therefore, this paper’s search for methods and strategies related to this topic was conducted in the following steps:
(1) We detail the specific databases and search engines (e.g., Science Direct, Google Scholar, Web of Science, CNKI) used, along with the keywords and search terms related to green development in the transportation industry, such as “green efficiency in transportation,” “carbon emissions in transport,” and “sustainable transportation pathways.”
(2) We specify the criteria used to select papers for our review, including the time frame of publication, relevance to the three main topics (green efficiency assessment, factors influencing green efficiency including carbon emissions, and analysis of future pathways for energy saving and emission reduction).
(3) We describe the process of screening the initially identified papers, assessing their relevance, and then selecting the most pertinent ones for detailed review.
We hope that these modifications satisfactorily address your concerns.
Comments 3:
The contribution of the paper is not clear crystal.
Response 3:
Agree. In response to your comment, we have revised the manuscript to more clearly articulate the contributions of our work. Specifically, we have made the following changes:
(1) We have revised the introduction to clearly outline the key contributions of our paper right at the beginning.
(2) The conclusion also has been restructured to succinctly summarize the main contributions, ensuring they are presented clearly and memorably.
In the introduction, revise the research purpose of this paper as follows:
“This study aims to review the current state of research on green development in the transportation sector and deeply explore the relationship between transportation systems and the economic-social-environmental systems. Utilizing research contents such as efficiency measurement, factor identification, and emission reduction pathway analysis, this study explores the operational and developmental potential of national or regional transportation systems. Simultaneously, it reveals the gaps and contradictions in existing research, providing direction for stakeholders seeking sustainable development pathways in the transportation industry and supporting the development decisions of governments and managers.”
In the conclusion, revise the contribution of this paper as follows:
“This study reviews the current state of research on green development in the transportation sector from three perspectives: development performance assessment, analysis of influencing factors, and exploration of development pathways. This systematic review provides a comprehensive framework for understanding sustainability issues in the transportation sector, offering robust support for government and managerial decision-making. Simultaneously, by revealing the gaps and contradictions in existing research, this paper provides clear guidance for the direction and focus of subsequent studies, further deepening and refining knowledge in this field.”
Comments 4:
The conclusion and findings are not accurate enough to be suggested to other researchers.
Response 4:
We agree with this comment. Therefore, based on the re-evaluation and consideration, we have revised our conclusions and findings to more accurately reflect the data and analysis. The newly revised conclusions are as follows:
“5. Conclusions
Excessive emissions and excessive energy consumption have heightened the global awareness of the necessity to strengthen green development in multiple countries. Green development in the transportation sector aims to reduce operational energy consumption in industrial development and control associated carbon emissions. Within the burgeoning actions for green industrial development, the transportation sector has become a primary target for improvement. Although some countries, led by China, still have a significant impact on the environment through their transportation sector, this sector has been able to identify crucial factors influencing its green development through the use of reasonable environmental performance measurement methods. Utilizing scenario analysis and other simulation methods, it explores future paths for energy conservation and emission reduction.
As the environmental impact of the transportation industry on issues such as carbon emissions and energy consumption becomes increasingly evident with its development, researchers, governments, and stakeholders have shown growing interest in understanding the mechanisms influencing green development in the transportation sector. This study reviews the current state of research on green development in the transportation sector from three perspectives: development performance assessment, analysis of influencing factors, and exploration of development pathways. This systematic review provides a comprehensive framework for understanding sustainability issues in the transportation sector, offering robust support for government and managerial decision-making. Simultaneously, by revealing the gaps and contradictions in existing research, this paper provides clear guidance for the direction and focus of subsequent studies, further deepening and refining knowledge in this field. Specifically, the conclusions of this study include the following aspects:
Firstly, this study presents the widely recognized input-output analysis framework in the transportation sector, focusing on the estimation methods for unintended outputs like carbon emissions and energy-environmental efficiency. However, there is a lack of internationally accepted methodologies for calculating transportation carbon emissions specific to different urban road design plans, modes of transport, and transportation vehicles. Environmental benefit assessments in the transportation industry predominantly measure direct energy consumption and carbon emissions during the developmental process, overlooking indirect energy consumption and pollutant emissions that could arise from inter-industrial interactions within national economic development. For example, carbon emissions embedded in intermediate products and services consumed by the industry and the inter-industry carbon transfer. Hence, the total carbon emissions and efficiency measurement results are not sufficiently accurate, which directly impacts the setting of carbon intensity targets. Moreover, in the evaluation of green development effects in the transportation sector, the literature based on static analyses at the national level or dynamic analyses combining provincial and municipal geographical locations is not entirely reliable. Considering spatial correlations, the geographical spatial dependency characteristics of green development should be incorporated into the research framework.
Secondly, in the realm of researching factors influencing the green development of the transportation sector, a systematic compilation has been conducted encompassing the aspects of influencing factors, the extent of influence, and the pathways of impact. However, due to the influence of the energy rebound effect and environmental regulations, the development directions of industrial energy consumption, pollutant emissions, and environmental benefits are not fixed. With the increasing prominence of technological levels as the core driver of the industry's green transformation and the growing demand of residents for the intelligence and greening of the transportation sector, research should intensify the analysis of the impact of technological innovation levels and resident preferences on advancing the greening process of the transportation sector.
Thirdly, as the internal structure of the transportation energy consumption and carbon emission systems is dissected, exploring the degree of influence of various factors, and simulating the improvement path of energy saving and emission reduction. The optimization of industrial structure, technological innovation, and economic scale have become more significant in the green development of the transportation industry. However, the most efficient means of reducing carbon emissions remains uncertain. Low-carbon travel options for residents can rapidly reduce carbon dioxide emissions at extremely low or even negative costs. Whether accelerating the development of intelligent transportation through source planning or leveraging technological advancements to empower the transformation of transportation energy structures, carbon reduction activities in the transportation sector largely depend on residents’ decisions for green travel. Coordinating consumer preferences with new technology applications, implementing green development concepts and requirements in the transportation industry, enhancing the quality and efficiency of transportation development, and optimizing the development layout of the transportation industry should be further explored.
Finally, by establishing scenarios that align with the developmental needs of the transportation sector, past research has achieved simulation analyses of the internal structure of the transportation energy consumption and emission system under the influence of transportation management policies and environmental constraints. Low-carbon scenarios, which incorporate conditions such as structural optimization, technological innovation, and policy constraints, have indeed resulted in lower carbon emissions and earlier achievement of carbon peak. However, from a life-cycle perspective, the lifetime mileage of each transportation system is seldom incorporated into scenario analysis studies, leaving a knowledge gap in scenario simulations for sustainable development plans for different vehicle types. Current scenario settings based on baseline, high-carbon, and low-carbon are still somewhat rudimentary. More specific parameter settings should be considered to formulate more scientifically-based emission reduction pathway enhancement schemes for the transportation sector, based on effective policy combinations. Additionally, scenario analyses mainly rely on predictions of energy consumption or carbon emissions. Comparing environmental performance or green total factor productivity across different scenario models could provide more valuable information for the future pursuit of intelligence and sustainable development in the transportation sector.”
The revised manuscript, with these changes, has been resubmitted for your review. We believe that these modifications address your concerns regarding the accuracy of our conclusions and findings, enhancing the overall quality and reliability of our paper.
We are grateful for your valuable input, which has significantly contributed to the improvement of our work.
Comments 5:
Some of the figures are not clear to the reader. please revise them.
Response 5:
Thank you for pointing this out. We agree with this comment. Therefore, we have increased the resolution of all figures to ensure they are clear and easily interpretable, even when zoomed in. We have adjusted the color schemes of the figures for better contrast and visibility, especially for colorblind readers, ensuring that the figures are accessible to a wider audience. Revised figures are in the newly submitted manuscript.

Reviewer 2 Report
Comments and Suggestions for Authors
Congratulations on the choice of subject matter. The layout of the manuscript is correct. The literature cited is relevant to the topic undertaken. I believe that the subject matter is original and important from the point of view of the development of science, although in my opinion, the following elements need to be clarified:
1. There are two “Table nr 3” in the article - this needs sorting out.
2. The purpose of the article needs to be clarified. Now, the stated purpose is too ambiguous.
3. The titles of Sections 3.0 and 3.2 are identical.
4. Line 504 „typical countries”: What did the author mean by that?
5. The research methodology was presented only laconically, and needs to be clarified.
6. It would be advisable to include information on the necessity of the research undertaken.
7. The lack of a clearly formulated research hypothesis. Therefore, I could not find information on whether the hypotheses were verified positively or negatively.
8. In “Conclusion”: no indication of the application value of the research, both in the theoretical and practical parts
9. The bibliography, there is no balance between Chinese and non-Chinese authors.
Author Response
Thank you for dedicating your valuable time to review our survey paper and for providing insightful feedback. We greatly value your comments and have thoroughly revised our manuscript in accordance with your suggestions.
For each point of feedback you raised, we have carefully considered and made corresponding revisions in the manuscript. These changes have been clearly marked in the resubmitted document with track changes for your ease of review.
In revising our paper, we have strived to ensure all information and viewpoints are up-to-date and accurate, while also improving the logical flow and readability of the text.
We look forward to your further feedback and hope that our revisions meet your expectations. Once again, thank you for your contribution and guidance towards improving our work.
Point-by-point response to Comments and Suggestions for Authors
Comments 1:
There are two “Table nr 3” in the article - this needs sorting out.
Response 1:
Agree. We have revised the title of different tables, which is in the page 7 and 10.
Comments 2:
The purpose of the article needs to be clarified. Now, the stated purpose is too ambiguous.
Response 2:
Thank you for your comment, we agree with your opinion and we have revised the manuscript to more clearly articulate the purpose of the article. Specifically, we have made the following changes:
(1) We have revised the introduction and abstract to clearly outline the key contributions of our paper right at the beginning.
(2) The conclusion has been restructured to succinctly summarize the main contributions, ensuring they are presented clearly and memorably.
The revised research purpose of this paper is:
“This study aims to review the current state of research on green development in the transportation sector and deeply explore the relationship between transportation systems and the economic-social-environmental systems. Utilizing research contents such as efficiency measurement, factor identification, and emission reduction pathway analysis, this study explores the operational and developmental potential of national or regional transportation systems. Simultaneously, it reveals the gaps and contradictions in existing research, providing direction for stakeholders seeking sustainable development pathways in the transportation industry and supporting the development decisions of governments and managers.” It is in the introduction section in page 3.
And “This study reviews the current state of research on green development in the transportation sector from three perspectives: development performance assessment, analysis of influencing factors, and exploration of development pathways. This systematic review provides a comprehensive framework for understanding sustainability issues in the transportation sector, offering robust support for government and managerial decision-making. Simultaneously, by revealing the gaps and contradictions in existing research, this paper provides clear guidance for the direction and focus of subsequent studies, further deepening and refining knowledge in this field.” It is in the conclusion section in page 18.
Comments 3:
The titles of Sections 3.0 and 3.2 are identical.
Response 3:
In response to your comment, we have revised the titles of Sections 3.0 and 3.2 as follows:
“3. Research on the mechanism of green development in the transportation industry
3.2. Analysis of the impact pathways for green development in the transportation industry”
Comments 4:
Line 504 „typical countries”: What did the author mean by that?
Response 4:
Thank you for your question regarding the term “typical countries” used in line 504 of our manuscript. By “typical countries,” we meant to refer to countries that are representative examples in the context of our study’s focus. Specifically, in this instance, we are referring to countries like China, the United States, Germany, and the United Kingdom, which are notable for their significant role in global energy consumption and economic growth.
These countries were selected as examples because they have substantial influence in the global energy market, both as major consumers and producers of fossil fuels, and are at the forefront of the transition towards greener and cleaner energy production. Their policies and initiatives in energy and transportation sectors are often indicative of broader trends and challenges in the shift towards sustainable energy use globally.
We have revised the manuscript to clarify this point and ensure that the term “typical countries” is accurately understood in the context of our research focus. The revision now reads as follows:
" Some typical countries, such as China, the United States, Germany, and the United Kingdom, still persist in enhancing their economic growth by utilizing non-renewable energy resources based on fossil fuels. The pace at which these economies are moving towards greener or cleaner energy production is far greater than other global energy users. Therefore, this study focuses on analyzing the impact of these countries’ policies on the green development of the transportation sector. Table 5 reveals specific policy measures taken by regions with a high share of transportation sector emissions, led by the United States, to achieve green development goals in transportation.", which is also shown in Line 492-500.
Comments 5:
The research methodology was presented only laconically, and needs to be clarified.
Response 5:
Thank you for your feedback regarding the presentation of our research methodology. We understand the importance of clearly articulating our methodological approach to ensure the robustness and reproducibility of our study.
In response to your comments, we have taken the following steps to enhance the clarity of the methodology section:
(1) We have revised the methodology section to ensure clarity and ease of understanding. We recognize that the technical details were dense and might have impeded comprehension. Therefore, we have simplified the language, ensuring that the reader can easily follow the progression and rationale behind our methodological choices.
(2) To better contextualize the relevance of the selected methodologies to our study, we have added a detailed explanation of how each methodology specifically applies to the transportation sector and the particular issues we are addressing. This includes discussing the suitability of these methodologies for analyzing sustainable development efficiency within this sector and how they help in addressing the research questions.
(3) We have thoroughly reviewed our citations and references related to the methodologies. Wherever methodologies are mentioned, we have now included citations to the foundational studies or authors that developed these methods.
The contents including these improvements are located on pages 5-6 and page 14-17 of the revised manuscript:
“2.2. Methods for measuring green development efficiency
2.2.1 Static Analysis Methodology
The essence of green efficiency lies in the consideration of pollution emissions and energy consumption within the efficiency of production technology. As a form of relative efficiency analysis, the measurement of green efficiency necessitates the construction of a production frontier, hence, Stochastic Frontier Analysis (SFA) and Data Envelopment Analysis (DEA) are commonly employed assessment methods. The SFA model, which predetermines the form of the production function, is specifically applied to the production processes of enterprises. It serves as an efficient means to eliminate the impact of managerial efficiency in measuring the efficiency of various indicators. However, it is susceptible to structural biases due to potential mis-specification of the production function [24-26]. DEA model, capable of better fitting multi-output production activities that include undesired outputs and circumventing the rigid assumptions of model specification and normal distribution of stochastic error terms inherent in SFA, finds more extensive application in efficiency evaluation from a static perspective [27]. Utilizing linear programming and convex analysis to establish the production frontier boundary, the DEA model projects different Decision-Making Units (DMUs) onto this boundary. The relative efficiency among DMUs is evaluated based on their deviation from this frontier, where DMUs on the boundary are considered technically efficient (efficiency=1), while those below it is deemed technically inefficient (efficiency<1).
Moreover, a fundamental requirement of the traditional DEA model is the minimization of inputs for a corresponding maximization of outputs, making it unsuitable to incorporate environmental pollution variables. Hence, numerous methodologies have been proposed to integrate environmental pollution variables into the productivity analysis framework. For instance, multi-stage DEA models [28, 29], SBM-DEA models [30-32], Super-SBM models [33, 34], and network DEA models [35, 36] consider the impacts of random errors and relaxation of factors. By distinguishing environmental factors, random errors, and internal management variables, these extended models enhance the accuracy of measuring green development efficiency in industries such as manufacturing and transportation. However, DEA models do not account for temporal factors and efficiency changes over time, which may lead to incompleteness in evaluation results under certain circumstances.
2.2.2 Dynamic Analysis Methodology
The DEA model is limited in its ability to reflect the trend of productivity changes and often requires a long time series of empirical data to deduce the dynamic characteristics of efficiency. Thus, the Malmquist index based on directional distance functions [37-39], and its extended analytical methods such as the Meta-frontier Malmquist index analysis [40,41], have been developed to overcome the lack of dynamic perspective in efficiency evaluation inherent in the DEA model. These methods are pivotal in assessing the environmental performance of the transportation sector from a dynamic angle. Additionally, the actual production technology efficiency in the industry encompasses a wide array of variables, many of which are beyond subjective control. Research has introduced external environmental variables such as technological advancement, environmental regulations, and asset structure, continuously refining the green development evaluation indicators for the transportation industry [42-44]. By synthesizing indicator selections and quantification methods from domestic and international literature, the selection and quantification methods for input, desired output, and undesired output indicators have gained widespread recognition. Detailed selections and quantification methods of these indicators are presented in Table 2.
2.2.3 Spatial Analysis Methodology
With the progressively widespread application of spatial econometric methods, the spatial spillover effects of carbon emissions, a crucial indicator of industrial development efficiency, have garnered extensive attention. Utilizing spatial autocorrelation analysis to construct spatial panel models, it has been observed that technological externalities and production process dependencies in industrial development significantly influence carbon emissions due to characteristics of neighboring regions, exhibiting notable spatial correlations [45,46]. In the case of the transportation sector, a vital inter-regional connector, its spatial dependency is even more pronounced. Numerous studies have confirmed the spatial clustering characteristics and regional disparities in carbon emissions from the transportation sector, with economically advanced regions being more affected [47-49]. Furthermore, to explore the structural characteristics of total carbon emissions after integrating spatial correlation effects, as well as to delve deeper into the spatial patterns and evolution of carbon emissions across different industries and sectors [50], social network analysis has begun to be applied in studying the spatial correlation networks of carbon emissions. Many scholars have employed this methodology to construct networks such as aviation [51,52], urban public transportation [53,54], industry-wide mobility [55], and regional carbon emission networks [56,57], to simulate and analyze the spatial evolution of carbon emissions.”
And “4.2. Prediction Methodology
For the design and implementation of green development in urban transportation, understanding and analyzing the interactions among a range of dynamic factors that shape transportation patterns, behaviors, and impacts is crucial. System Dynamics models (SD) facilitate in-depth studies of the complex system composed of transportation, socio-economic, energy, and environmental components, and simulate predictions of the effectiveness of green development strategies implemented in the transportation sector [161-163]. SD models are composed of stocks, flows, and auxiliary variables, used to analyze complex dynamic feedback systems. They graphically ex-press the interactions among various factors in the transportation carbon emission system through stock-flow diagrams [164-166]. Owing to their proficiency in handling nonlinear, high-order, multivariate, multi-feedback, and cyclical system issues, System Dynamics methods are widely applied in studies on energy conservation and carbon emissions, including in socio-economics, primary, secondary, and tertiary industries, housing, transportation, waste management, and electricity [167,168]. The transportation industry's carbon emission system is a complex system with dynamic changes, typically divided into economic, energy, and environmental subsystems, or road, rail, and waterway subsystems. Consolidating past research, Figures 12 and 13 respectively represent the causal loop diagram and system flow diagram of the complex transportation-emission system, revealing the paths of energy consumption and pollution emissions in the transportation sector. Variables in the system can be categorized based on their attributes and significance into auxiliary, rate, and level variables. Auxiliary variables represent relationships between main variables, such as energy consumption in the transportation industry and environmental regulatory policies. Rate variables refer to the inputs and outputs of level variables, indicating changes in level variables over unit time, such as the share of the tertiary industry, and GDP growth rate. On the other hand, state variables represent significant stock variables, like GDP, average annual population, and carbon emissions. Ultimately, governments can simulate energy-saving and emission-reduction pathways in the transportation sector and predict pollution emission outcomes under current scenarios by combining local environmental protection and green industrial development policies, utilizing parameter-setting results [169-171].
The Long-range Energy Alternatives Planning System (LEAP) models energy supply, utilization, and conversion technologies by modeling specific technical and economic parameters. Given its capability to quantitatively analyze the dynamic impacts of energy policies on the environment and forecast long-term energy demands and pollution emissions, LEAP is widely applied across various scales, from national to regional and industrial levels, particularly in sectors like industry [172,173], transportation [174,175], and commerce [176]. Focusing on energy and carbon emissions, research primarily concentrates on future emission reduction potentials and pathways towards low-carbon transformation, emphasizing the significant impact of enhancing energy efficiency and changing energy structures on regional green development. Although the LEAP model effectively simulates energy consumption situations and their environmental impacts, it falls short of fully capturing the potential socio-economic benefits. The notable impacts of cross-industry energy transfer and emerging energy transformations on traditional development models, especially in developing countries, require further investigation [177]. Based on the scale, structural, technological effects, and external constraint policies impacting the green development of the transportation sector, this study, drawing on a review of relevant literature, proposes three scenarios (baseline, low-carbon, and high-carbon), as shown in Table 6. The low-carbon scenario includes sub-scenarios like structural optimization, technological progress, environmental policy constraints, and coordinated economic-environmental development. Given that industrial investment scale, energy intensity, energy structure, intensity of technological investments, and environmental regulation indicators play varying roles in different scenarios, predictions of carbon emissions and policy implementation outcomes inevitably differ. Building on the research by Yang et al. (2021) [178] and IEA data, Figure 15 illustrates the carbon emission outcomes predicted using the LEAP model under different scenarios. In the low-carbon scenario for the transportation sector, the growth and peak of carbon emissions are consistently lower than other scenarios, more in line with energy conservation and emission reduction policy requirements. Therefore, for high-pollution industries like transportation, optimizing industrial structures, enhancing technological innovation levels, and formulating reasonable environmental regulatory policies are key to promoting coordinated development between industry and environment and achieving a green development path [179-181].”
Comments 6:
It would be advisable to include information on the necessity of the research undertaken.
Response 6:
Thank you for your constructive suggestion to include information on the necessity of the research undertaken in our manuscript. We agree that highlighting the importance and relevance of our research is crucial for readers to understand the context and significance of our study. In response to your feedback, we have made the following revisions to our manuscript:
(1) We have revised the introduction to include a more detailed explanation of the necessity of our research. This includes discussing the current gaps in the literature, the specific challenges or problems our research addresses, and the potential impact of our findings on the field.
(2) In the conclusion section, we have elaborated on the practical implications of our research, demonstrating how our findings can be applied or used in real-world settings or contribute to policy or practice.
(3) We have made it clear how our research adds value to the field, fills existing gaps, and addresses unanswered questions, thereby underscoring its necessity.
The following content is the revised introduction section, which is in page 1 to 3:
“1. Introduction
In the current scenario where transportation predominantly contributes to extreme pollution, energy-consuming nations are continuously striving to find ways to ensure environmental sustainability. The transportation sector, a critical link between a nation’s production and consumption, is also one of the primary sources of energy consumption and carbon emissions. Since 2017, the transportation sector has emerged as the world's second-largest contributor. Despite a decrease in residents’ travel needs due to the significant global public health event in 2020, energy consumption in the transportation field still accounted for about 26% of the total in 2022 [1], and carbon emissions amounted to approximately 21% [2]. Projections by the IEA suggest that by 2030, the share of CO2 emissions from the transportation sector might rise to 50%, and by 2050, it is expected to reach 80% (as shown in Figure 1). Experiences from developed countries indicate that only with the transportation sector’s synchronous development with the economy can the overall advantages and comprehensive benefits of the transportation system be fully realized, elevating the level of transportation development to new heights [3]. Simultaneously, the ‘Green Economy Blue Book’ points out that ‘green development’ represents a more resource-efficient, cleaner, and recoverable state of development, an active interaction between ‘economy-nature-society’ [4], and a state of balance between resources, environment, and economic development [5]. With the rapid growth of emerging industries such as ride-hailing services, shared bicycles, and online freight platforms, low-carbon production, and lifestyles in the transportation sector is gradually taking shape. The concept of green development has permeated the transportation industry, altering the direction of its development. The transportation sector has begun to focus more on the quality and efficiency of development, rather than speed and scale. A search in CNKI and Science Direct using relevant keywords reveals a significant increasing trend in research on green development in the transportation field across various countries (as shown in Figure 2), yet there is still vast room for research.
The global transportation industry continues to face immense pressure to reduce emissions, particularly those of energy consumption and carbon emissions. Figure 3 shows the variations in CO2 emissions from the transportation sectors of 18 typical economies [1] [6-8]. Represented by the United States, France, Germany, Portugal, and the United Kingdom, the CO2 emissions from the transportation sectors of many countries occupy a relatively large share of their total energy consumption carbon emissions, especially in the United States, where it accounts for 33% of national carbon emissions. Meanwhile, for many EU countries in the middle to late stages of industrialization, with relatively stable transportation service development, the transportation sector has become the only sector with continuous growth in carbon emissions. Therefore, the development of the global high-energy-consuming industry - the transportation sector - should adapt to various urbanization needs and simultaneously reduce its negative impact on the environment to achieve ecological sustainability [9]. Global research has progressively confirmed the correlation between transportation and green development, with an increasing number of studies emphasizing the construction of more comprehensive and integrated approaches to understand the significance of achieving green development in the transportation sector.

Reviewer 3 Report
Comments and Suggestions for Authors
The article discusses the green development of the transportation industry, emphasizing the need for diverse measurement methods for carbon emissions, a deeper understanding of the mechanisms driving green development, and consideration of residents' choices in reducing emissions. It also points out the lack of comprehensive research on the impact of intelligent transportation systems and suggests further exploration in optimizing green development through effective policies and emission reduction plans. Overall the work is scientifically sound, but English use is difficult to understand and deserves a thorough correction of style to ease its reading and enhance the key points of the research. For instance, expressions such as "Considering the endogenization of the level of technology, the influence of the interaction between the influencing factors, etc. on the degree and direction of the role of the factors..." make no sense and need a second writing.
Comments on the Quality of English LanguageThe article discusses the green development of the transportation industry, emphasizing the need for diverse measurement methods for carbon emissions, a deeper understanding of the mechanisms driving green development, and consideration of residents' choices in reducing emissions. It also points out the lack of comprehensive research on the impact of intelligent transportation systems and suggests further exploration in optimizing green development through effective policies and emission reduction plans. Overall the work is scientifically sound, but English use is difficult to understand and deserves a thorough correction of style to ease its reading and enhance the key points of the research. For instance, expressions such as "Considering the endogenization of the level of technology, the influence of the interaction between the influencing factors, etc. on the degree and direction of the role of the factors..." make no sense and need a second writing.
Author Response
Thank you for dedicating your valuable time to review our survey paper and for providing insightful feedback. We greatly value your comments and have thoroughly revised our manuscript in accordance with your suggestions.
For each point of feedback you raised, we have carefully considered and made corresponding revisions in the manuscript. These changes have been clearly marked in the resubmitted document with track changes for your ease of review.
In revising our paper, we have strived to ensure all information and viewpoints are up-to-date and accurate, while also improving the logical flow and readability of the text.
We look forward to your further feedback and hope that our revisions meet your expectations. Once again, thank you for your contribution and guidance towards improving our work.
Point-by-point Response to Comments and Suggestions for Authors
Comments 1:
Overall, the work is scientifically sound, but English use is difficult to understand and deserves a thorough correction of style to ease its reading and enhance the key points of the research. For instance, expressions such as “Considering the endogenization of the level of technology, the influence of the interaction between the influencing factors, etc. on the degree and direction of the role of the factors...” make no sense and need a second writing.
Response 1:
Thank you for acknowledging the scientific soundness of our work and for your valuable feedback regarding the use of English in our manuscript. We understand the importance of clear and comprehensible writing to effectively communicate our research findings and appreciate your specific example highlighting the need for improvement. Therefore, we have undertaken the following steps:
(1) We have thoroughly reviewed and revised the manuscript.
(2) We are specifically revisiting and rewriting complex and unclear sentences, such as the one you pointed out. Our goal is to simplify the language while retaining the intended meaning, making it more accessible to readers.
(3) We are working to improve the overall clarity and flow of the manuscript by restructuring sentences, using more direct language, and avoiding jargon or overly complex expressions.
The revised manuscript, with improved language and style, has been resubmitted for your review. We believe these changes significantly enhance the readability of our work and effectively convey the key points of our research.
We hope that these revisions meet your expectations and greatly appreciate your assistance in improving the quality of our manuscript.
Thank you again for your constructive feedback.

Reviewer 4 Report
Comments and Suggestions for Authors
Dear authors, first I would like to express my gratitude for the opportunity to review your article. After a thorough analysis of tour article, I would like to provide some constructive suggestions to enhance the clarity and impact of your work:
1.Introduction
Reviewer: The introduction exhibits several notable shortcomings that need to be addressed to enhance the overall quality of the paper:
The introduction appears somewhat disorganized and challenging to follow. Key points and the article's structure lack clear delineation. I suggest improving the introduction's organization by explicitly highlighting the main topics to be addressed. This will render the content more accessible to readers and facilitate understanding.
There is an overuse of outdated data in the introduction. Specific data, such as carbon emission statistics from 2020, are mentioned without indicating whether these figures remain current or have been affected by recent events. I strongly recommend updating this data to reflect the most recent situation, unless there is a specific reason for using older data.
Furthermore, the introduction lacks citations. Several statements about statistics and information are made, but there are no references or citations to support these claims. This omission weakens the article's credibility. Ensure that proper sources are included to substantiate the presented information.
The introduction also lacks adequate global context. While the article focuses on carbon emissions in the transportation sector, it does not provide sufficient global context. Explain why this topic is relevant in a global context and how carbon emissions in the transportation sector compare to other sources.
Lastly, the introduction fails to clearly define the research objectives. I strongly recommend including a section that explicitly describes what you aim to achieve with the study. Defining the objectives will help readers understand the purpose of the work from the outset. Additionally, it's crucial to identify the literature gap that this study intends to fill. Why is this research necessary, and what is the original contribution of this work? This should be clearly communicated in the introduction. Furthermore, explain why this study is relevant. What is the practical importance, and what are the political, economic, or social implications of your work? Connecting the research problem with the real need to address it can enhance the attractiveness of the article.
2.2. Methods for measuring green development efficiency
Reviewer: The lack of clarity and theoretical grounding regarding the applied methodology is a significant weakness in this study. Therefore, I suggest the following:
Clarity in Presentation: The section starts well, introducing standard methodologies for assessing sustainable development efficiency. However, as the section progresses, it can become somewhat dense in terms of technical information. Ensure that the exposition is clear, and the reader can easily follow the different methodologies.
Contextualization: It is essential to contextualize why these methodologies are relevant to the study at hand. How do these methodologies apply to the transportation sector and the specific issue under discussion? This would help justify the choice of these methodologies.
Citations and References: When mentioning the different methodologies, it's crucial to cite the sources (e.g., studies or authors) that introduced or developed these methodologies. Make sure to include proper bibliographic references to support the validity and utility of these approaches. Consider studies such as (https://www.tandfonline.com/doi/abs/10.1080/1528008X.2022.2112807) to justify review methodological approaches.
5. Conclusions
Reviewer: I would like to begin by expressing my appreciation for your dedication to research and writing in the article on sustainable development in the transportation sector. It is evident that meticulous work was carried out in identifying the limitations of existing research, which is valuable to the academic community. However, I'd like to emphasize the importance of enriching the conclusion of the study with essential elements that are currently lacking.
The article's conclusion heavily focuses on exposing limitations, which is important, but it lacks emphasis on the significant conclusions of the study. It would be valuable to highlight the main findings that the study has provided. What insights were gained? What new knowledge was generated? These conclusions are at the core of the article's value.
Furthermore, it's important that the authors clearly identify which literature gaps have been addressed through this study. How does the article contribute to the current understanding? The authors should indicate how their research helps fill specific gaps and advance the field.
The conclusion can be enriched by discussing the theoretical and practical implications of the findings. How can the study influence existing theory? Additionally, what are the practical implications for professionals in the transportation sector, policymakers, and other stakeholders? Such discussions enhance the article's impact.
It is crucial that the conclusion highlights the unique contribution of this study to the existing literature. Why is this article important? What advancements does it bring? The authors should emphasize the relevance of their work to the academic and professional community. Consider creating a research agenda to add value to the article.
In addition to limitations, it is helpful to include a succinct summary of the main conclusions and the study's contribution, making the conclusion more accessible and easy to understand. Enriching the conclusion with these elements will ensure that the article not only identifies research limitations but also clearly communicates the findings and the authors' contribution to the field. This will make the article more comprehensive and impactful.
Comments on the Quality of English Language
The English language just required moderate editing.
Author Response
Thank you for dedicating your valuable time to review our survey paper and for providing insightful feedback. We greatly value your comments and have thoroughly revised our manuscript in accordance with your suggestions.
For each point of feedback you raised, we have carefully considered and made corresponding revisions in the manuscript. These changes have been clearly marked in the resubmitted document with track changes for your ease of review.
In revising our paper, we have strived to ensure all information and viewpoints are up-to-date and accurate, while also improving the logical flow and readability of the text.
We look forward to your further feedback and hope that our revisions meet your expectations. Once again, thank you for your contribution and guidance towards improving our work.
Point-by-point response to Comments and Suggestions for Authors
Comments 1:
1.Introduction
Reviewer: The introduction exhibits several notable shortcomings that need to be addressed to enhance the overall quality of the paper:
The introduction appears somewhat disorganized and challenging to follow. Key points and the article's structure lack clear delineation. I suggest improving the introduction's organization by explicitly highlighting the main topics to be addressed. This will render the content more accessible to readers and facilitate understanding.
There is an overuse of outdated data in the introduction. Specific data, such as carbon emission statistics from 2020, are mentioned without indicating whether these figures remain current or have been affected by recent events. I strongly recommend updating this data to reflect the most recent situation, unless there is a specific reason for using older data.
Furthermore, the introduction lacks citations. Several statements about statistics and information are made, but there are no references or citations to support these claims. This omission weakens the article's credibility. Ensure that proper sources are included to substantiate the presented information.
The introduction also lacks adequate global context. While the article focuses on carbon emissions in the transportation sector, it does not provide sufficient global context. Explain why this topic is relevant in a global context and how carbon emissions in the transportation sector compare to other sources.
Lastly, the introduction fails to clearly define the research objectives. I strongly recommend including a section that explicitly describes what you aim to achieve with the study. Defining the objectives will help readers understand the purpose of the work from the outset. Additionally, it's crucial to identify the literature gap that this study intends to fill. Why is this research necessary, and what is the original contribution of this work? This should be clearly communicated in the introduction. Furthermore, explain why this study is relevant. What is the practical importance, and what are the political, economic, or social implications of your work? Connecting the research problem with the real need to address it can enhance the attractiveness of the article.
Response 1:
Thank you for your detailed and constructive feedback on the introduction of our paper. We appreciate your insights and have taken the following steps to address each of the concerns you raised:
(1) To make the introduction more coherent and easier to follow, we have restructured it to clearly highlight the main topics. We now explicitly delineate the key points and the structure of the article in the introduction, ensuring a logical flow that facilitates reader comprehension.
(2) We acknowledge the importance of using current data and have updated our introduction with the latest statistics, including carbon emission figures up to 2022. This ensures that our paper reflects the most recent situation and remains relevant.
(3) We have thoroughly reviewed the introduction and added necessary citations to support our statements. This includes references for statistical data and information, thereby strengthening the credibility of our article.
(4) We have expanded the introduction to include a broader global context, especially regarding the negative global impact of carbon emissions from the transportation sector. This provides readers with a clearer understanding of the global relevance of our research topic.
Defining Research Objectives and Literature Gap: We have added a section that explicitly outlines the objectives of our study and the literature gap it intends to fill. This section clearly communicates the necessity and original contribution of our research, as well as its practical importance and wider implications. We believe this addition will greatly enhance the reader's understanding of the purpose and significance of our work.
The following content is the revised introduction section, which is in page 1 to 3:
“1. Introduction
In the current scenario where transportation predominantly contributes to extreme pollution, energy-consuming nations are continuously striving to find ways to ensure environmental sustainability. The transportation sector, a critical link between a nation’s production and consumption, is also one of the primary sources of energy consumption and carbon emissions. Since 2017, the transportation sector has emerged as the world's second-largest contributor. Despite a decrease in residents’ travel needs due to the significant global public health event in 2020, energy consumption in the transportation field still accounted for about 26% of the total in 2022 [1], and carbon emissions amounted to approximately 21% [2]. Projections by the IEA suggest that by 2030, the share of CO2 emissions from the transportation sector might rise to 50%, and by 2050, it is expected to reach 80% (as shown in Figure 1). Experiences from developed countries indicate that only with the transportation sector’s synchronous development with the economy can the overall advantages and comprehensive benefits of the transportation system be fully realized, elevating the level of transportation development to new heights [3]. Simultaneously, the ‘Green Economy Blue Book’ points out that ‘green development’ represents a more resource-efficient, cleaner, and recoverable state of development, an active interaction between ‘economy-nature-society’ [4], and a state of balance between resources, environment, and economic development [5]. With the rapid growth of emerging industries such as ride-hailing services, shared bicycles, and online freight platforms, low-carbon production and lifestyles in the transportation sector are gradually taking shape. The concept of green development has permeated the transportation industry, altering the direction of its development. The transportation sector has begun to focus more on the quality and efficiency of development, rather than speed and scale. A search in CNKI and Science Direct using relevant keywords reveals a significant increasing trend in research on green development in the transportation field across various countries (as shown in Figure 2), yet there is still vast room for research.
Figure 1. Global CO2 emission and projections by sector
Figure 2. Number of papers published in journals with relevant keywords from 2008 to 2022
The global transportation industry continues to face immense pressure to reduce emissions, particularly those of energy consumption and carbon emissions. Figure 3 shows the variations in CO2 emissions from the transportation sectors of 18 typical economies [1] [6-8]. Represented by the United States, France, Germany, Portugal, and the United Kingdom, the CO2 emissions from the transportation sectors of many countries occupy a relatively large share of their total energy consumption carbon emissions, especially in the United States, where it accounts for 33% of national carbon emissions. Meanwhile, for many EU countries in the middle to late stages of industrialization, with relatively stable transportation service development, the transportation sector has become the only sector with continuous growth in carbon emissions. Therefore, the development of the global high-energy-consuming industry - the transportation sector - should adapt to various urbanization needs and simultaneously reduce its negative impact on the environment to achieve ecological sustainability [9]. Global research has progressively confirmed the correlation between transportation and green development, with an increasing number of studies emphasizing the construction of more comprehensive and integrated approaches to understand the significance of achieving green development in the transportation sector.
Figure 3. The proportion of CO2 emissions from different industries in typical economies
In addressing the issue of green development in the transportation sector, research can be categorized into three pivotal aspects: ‘What’, ‘Why’, and ‘How’. Firstly, the quantification of green development indicators in the transportation sector has shifted from merely focusing on production, organizational, and service efficiencies to a more comprehensive measurement of green efficiency [10,11]. This primarily emphasizes energy consumption as an input indicator and direct carbon emissions as an undesired output indicator. However, the carbon transfer between the transportation sector and other sectors, given its role as a vital connector across regions and industries, is often overlooked. A contemplative approach is needed to scientifically measure the green development of transportation. Secondly, existing research has explored the key factors influencing the green development process of the transportation sector from multiple dimensions, including scale effect, structural effect, technological effect, and external policies [12,13]. However, the analysis of the correlation between technological innovation pace, market acceptance, energy consumption, and carbon emission levels is not thorough, influenced by the variability of short-term and long-term factors. Additionally, as the influence of resident preferences on urban transportation greening deepens, discussions of subjective factors should not be neglected. Lastly, to enhance green efficiency in the transportation sector and reinforce the importance of factors positively impacting energy conservation and emission reduction targets, the potential for carbon emission reduction in the transportation sector has garnered widespread attention among scholars globally [14]. Although various scenario simulations are increasingly refined, discussions from a full lifecycle perspective remain insufficient, and potential analyses based solely on carbon emissions lack persuasiveness.
This study aims to review the current state of research on green development in the transportation sector and deeply explore the relationship between transportation systems and the economic-social-environmental systems. Utilizing research contents such as efficiency measurement, factor identification, and emission reduction pathway analysis, this study explores the operational and developmental potential of national or regional transportation systems. Simultaneously, it reveals the gaps and contradictions in existing research, providing direction for stakeholders seeking sustainable development pathways in the transportation industry and supporting the development decisions of governments and managers.”
The revised introduction, along with these enhancements, also has been included in the resubmitted manuscript. We hope that these changes address your concerns and significantly improve the quality of our paper.
Thank you once again for your invaluable feedback, which has been instrumental in refining our manuscript.
Comments 2:
2.2. Methods for measuring green development efficiency
Reviewer: The lack of clarity and theoretical grounding regarding the applied methodology is a significant weakness in this study. Therefore, I suggest the following:
Clarity in Presentation: The section starts well, introducing standard methodologies for assessing sustainable development efficiency. However, as the section progresses, it can become somewhat dense in terms of technical information. Ensure that the exposition is clear, and the reader can easily follow the different methodologies.
Contextualization: It is essential to contextualize why these methodologies are relevant to the study at hand. How do these methodologies apply to the transportation sector and the specific issue under discussion? This would help justify the choice of these methodologies.
Citations and References: When mentioning the different methodologies, it's crucial to cite the sources (e.g., studies or authors) that introduced or developed these methodologies. Make sure to include proper bibliographic references to support the validity and utility of these approaches. Consider studies such as (https://www.tandfonline.com/doi/abs/10.1080/1528008X.2022.2112807) to justify review methodological approaches.
Response 2:
Thank you for your valuable feedback regarding the clarity and theoretical grounding of the methodology section in our study. We acknowledge the importance of these aspects and have made the following revisions to address your concerns:
(1) We have revised the methodology section to ensure clarity and ease of understanding. We recognize that the technical details were dense and might have impeded comprehension. Therefore, we have simplified the language, ensuring that the reader can easily follow the progression and rationale behind our methodological choices.
(2) To better contextualize the relevance of the selected methodologies to our study, we have added a detailed explanation of how each methodology specifically applies to the transportation sector and the particular issues we are addressing. This includes discussing the suitability of these methodologies for analyzing sustainable development efficiency within this sector and how they help in addressing the research questions.
(3) We have thoroughly reviewed our citations and references related to the methodologies. Wherever methodologies are mentioned, we have now included citations to the foundational studies or authors that developed these methods.
The contents including these improvements are located on pages 5 and 6 of the revised manuscript:
“2.2. Methods for measuring green development efficiency
2.2.1 Static Analysis Methodology
The essence of green efficiency lies in the consideration of pollution emissions and energy consumption within the efficiency of production technology. As a form of relative efficiency analysis, the measurement of green efficiency necessitates the construction of a production frontier, hence, Stochastic Frontier Analysis (SFA) and Data Envelopment Analysis (DEA) are commonly employed assessment methods. The SFA model, which predetermines the form of the production function, is specifically applied to the production processes of enterprises. It serves as an efficient means to eliminate the impact of managerial efficiency in measuring the efficiency of various indicators. However, it is susceptible to structural biases due to potential mis-specification of the production function [24-26]. DEA model, capable of better fitting multi-output production activities that include undesired outputs and circumventing the rigid assumptions of model specification and normal distribution of stochastic error terms inherent in SFA, finds more extensive application in efficiency evaluation from a static perspective [27]. Utilizing linear programming and convex analysis to establish the production frontier boundary, the DEA model projects different Decision-Making Units (DMUs) onto this boundary. The relative efficiency among DMUs is evaluated based on their deviation from this frontier, where DMUs on the boundary are considered technically efficient (efficiency=1), while those below it is deemed technically inefficient (efficiency<1).
Moreover, a fundamental requirement of the traditional DEA model is the minimization of inputs for a corresponding maximization of outputs, making it unsuitable to incorporate environmental pollution variables. Hence, numerous methodologies have been proposed to integrate environmental pollution variables into the productivity analysis framework. For instance, multi-stage DEA models [28, 29], SBM-DEA models [30-32], Super-SBM models [33, 34], and network DEA models [35, 36] consider the impacts of random errors and relaxation of factors. By distinguishing environmental factors, random errors, and internal management variables, these extended models enhance the accuracy of measuring the green development efficiency in industries such as manufacturing and transportation. However, DEA models do not account for temporal factors and efficiency changes over time, which may lead to incompleteness in evaluation results under certain circumstances.
2.2.2 Dynamic Analysis Methodology
The DEA model is limited in its ability to reflect the trend of productivity changes and often requires a long time series of empirical data to deduce the dynamic characteristics of efficiency. Thus, the Malmquist index based on directional distance functions [37-39], and its extended analytical methods such as the Meta-frontier Malmquist index analysis [40,41], have been developed to overcome the lack of dynamic perspective in efficiency evaluation inherent in the DEA model. These methods are pivotal in assessing the environmental performance of the transportation sector from a dynamic angle. Additionally, the actual production technology efficiency in the industry encompasses a wide array of variables, many of which are beyond subjective control. Research has introduced external environmental variables such as technological advancement, environmental regulations, and asset structure, continuously refining the green development evaluation indicators for the transportation industry [42-44]. By synthesizing indicator selections and quantification methods from domestic and international literature, the selection and quantification methods for input, desired output, and undesired output indicators have gained widespread recognition. Detailed selections and quantification methods of these indicators are presented in Table 2.
2.2.3 Spatial Analysis Methodology
With the progressively widespread application of spatial econometric methods, the spatial spillover effects of carbon emissions, a crucial indicator of industrial development efficiency, have garnered extensive attention. Utilizing spatial autocorrelation analysis to construct spatial panel models, it has been observed that technological externalities and production process dependencies in industrial development significantly influence carbon emissions due to characteristics of neighboring regions, exhibiting notable spatial correlations [45,46]. In the case of the transportation sector, a vital inter-regional connector, its spatial dependency is even more pronounced. Numerous studies have confirmed the spatial clustering characteristics and regional disparities in carbon emissions from the transportation sector, with economically advanced regions being more affected [47-49]. Furthermore, to explore the structural characteristics of total carbon emissions after integrating spatial correlation effects, as well as to delve deeper into the spatial patterns and evolution of carbon emissions across different industries and sectors [50], social network analysis has begun to be applied in studying the spatial correlation networks of carbon emissions. Many scholars have employed this methodology to construct networks such as aviation [51,52], urban public transportation [53,54], industry-wide mobility [55], and regional carbon emission networks [56,57], to simulate and analyze the spatial evolution of carbon emissions.”
We believe that these changes significantly improve the clarity and theoretical grounding of the methodology section, making it more accessible and relevant to our research context.
Comments 3:
- Conclusions
Reviewer: I would like to begin by expressing my appreciation for your dedication to research and writing in the article on sustainable development in the transportation sector. It is evident that meticulous work was carried out in identifying the limitations of existing research, which is valuable to the academic community. However, we’d like to emphasize the importance of enriching the conclusion of the study with essential elements that are currently lacking.
The article’s conclusion heavily focuses on exposing limitations, which is important, but it lacks emphasis on the significant conclusions of the study. It would be valuable to highlight the main findings that the study has provided. What insights were gained? What new knowledge was generated? These conclusions are at the core of the article's value.
Furthermore, it’s important that the authors clearly identify which literature gaps have been addressed through this study. How does the article contribute to the current understanding? The authors should indicate how their research helps fill specific gaps and advance the field.
The conclusion can be enriched by discussing the theoretical and practical implications of the findings. How can the study influence existing theory? Additionally, what are the practical implications for professionals in the transportation sector, policymakers, and other stakeholders? Such discussions enhance the article's impact.
It is crucial that the conclusion highlights the unique contribution of this study to the existing literature. Why is this article important? What advancements does it bring? The authors should emphasize the relevance of their work to the academic and professional community. Consider creating a research agenda to add value to the article.
In addition to limitations, it is helpful to include a succinct summary of the main conclusions and the study's contribution, making the conclusion more accessible and easy to understand. Enriching the conclusion with these elements will ensure that the article not only identifies research limitations but also clearly communicates the findings and the authors' contribution to the field. This will make the article more comprehensive and impactful.
Response 3:
We agree with this comment. Therefore, based on the re-evaluation and consideration, we have revised our conclusions and findings to more accurately reflect the data and analysis. The newly revised conclusions are as follows:
“5. Conclusions
Excessive emissions and excessive energy consumption have heightened the global awareness of the necessity to strengthen green development in multiple countries. Green development in the transportation sector aims to reduce operational energy consumption in industrial development and control associated carbon emissions. Within the burgeoning actions for green industrial development, the transportation sector has become a primary target for improvement. Although some countries, led by China, still have a significant impact on the environment through their transportation sector, this sector has been able to identify crucial factors influencing its green development through the use of reasonable environmental performance measurement methods. Utilizing scenario analysis and other simulation methods, it explores future paths for energy conservation and emission reduction.
As the environmental impact of the transportation industry on issues such as carbon emissions and energy consumption becomes increasingly evident with its development, researchers, governments, and stakeholders have shown growing interest in understanding the mechanisms influencing green development in the transportation sector. This study reviews the current state of research on green development in the transportation sector from three perspectives: development performance assessment, analysis of influencing factors, and exploration of development pathways. This systematic review provides a comprehensive framework for understanding sustainability issues in the transportation sector, offering robust support for government and managerial decision-making. Simultaneously, by revealing the gaps and contradictions in existing research, this paper provides clear guidance for the direction and focus of subsequent studies, further deepening and refining knowledge in this field. Specifically, the conclusions of this study include the following aspects:
Firstly, this study presents the widely recognized input-output analysis framework in the transportation sector, focusing on the estimation methods for unintended outputs like carbon emissions and energy-environmental efficiency. However, there is a lack of internationally accepted methodologies for calculating transportation carbon emissions specific to different urban road design plans, modes of transport, and transportation vehicles. Environmental benefit assessments in the transportation industry predominantly measure direct energy consumption and carbon emissions during the developmental process, overlooking indirect energy consumption and pollutant emissions that could arise from inter-industrial interactions within national economic development. For example, carbon emissions embedded in intermediate products and services consumed by the industry, and the inter-industry carbon transfer. Hence, the total carbon emissions and efficiency measurement results are not sufficiently accurate, which directly impacts the setting of carbon intensity targets. Moreover, in the evaluation of green development effects in the transportation sector, the literature based on static analyses at the national level or dynamic analyses combining provincial and municipal geographical locations is not entirely reliable. Considering spatial correlations, the geographical spatial dependency characteristics of green development should be incorporated into the research framework.
Secondly, in the realm of researching factors influencing the green development of the transportation sector, a systematic compilation has been conducted encompassing the aspects of influencing factors, the extent of influence, and the pathways of impact. However, due to the influence of the energy rebound effect and environmental regulations, the development directions of industrial energy consumption, pollutant emissions, and environmental benefits are not fixed. With the increasing prominence of technological levels as the core driver of the industry's green transformation and the growing demand of residents for the intelligence and greening of the transportation sector, research should intensify the analysis of the impact of technological innovation levels and resident preferences on advancing the greening process of the transportation sector.
Thirdly, as the internal structure of the transportation energy consumption and carbon emission systems is dissected, exploring the degree of influence of various factors, and simulating the improvement path of energy saving and emission reduction. The optimization of industrial structure, technological innovation, and economic scale have become more significant in the green development of the transportation industry. However, the most efficient means of reducing carbon emissions remains uncertain. Low-carbon travel options for residents can rapidly reduce carbon dioxide emissions at extremely low or even negative costs. Whether accelerating the development of intelligent transportation through source planning or leveraging technological advancements to empower the transformation of transportation energy structures, carbon reduction activities in the transportation sector largely depend on residents’ decisions for green travel. Coordinating consumer preferences with new technology applications, implementing green development concepts and requirements in the transportation industry, enhancing the quality and efficiency of transportation development, and optimizing the development layout of the transportation industry should be further explored.
Finally, by establishing scenarios that align with the developmental needs of the transportation sector, past research has achieved simulation analyses of the internal structure of the transportation energy consumption and emission system under the influence of transportation management policies and environmental constraints. Low-carbon scenarios, which incorporate conditions such as structural optimization, technological innovation, and policy constraints, have indeed resulted in lower carbon emissions and earlier achievement of carbon peak. However, from a life-cycle perspective, the lifetime mileage of each transportation system is seldom incorporated into scenario analysis studies, leaving a knowledge gap in scenario simulations for sustainable development plans for different vehicle types. Current scenario settings based on baseline, high-carbon, and low-carbon are still somewhat rudimentary. More specific parameter settings should be considered to formulate more scientifically-based emission reduction pathway enhancement schemes for the transportation sector, based on effective policy combinations. Additionally, scenario analyses mainly rely on predictions of energy consumption or carbon emissions. Comparing environmental performance or green total factor productivity across different scenario models could provide more valuable information for the future pursuit of intelligence and sustainable development in the transportation sector.”
The revised manuscript, with these changes, has been resubmitted for your review. We believe that these modifications address your concerns regarding the accuracy of our conclusions and findings, enhancing the overall quality and reliability of our paper.
We are grateful for your valuable input, which has significantly contributed to the improvement of our work.
Response to Comments on the Quality of English Language
Comments 1:
The English language just required moderate editing.
Response 1:
Thank you for your positive feedback on the overall quality of the English language used in our manuscript. We appreciate your suggestion regarding the necessity of moderate checking to further refine the text.
To address this, we have taken the following steps:
(1) We have conducted an additional thorough review of the manuscript to identify and correct any minor language issues. This includes a careful examination of grammar, punctuation, and phrasing to ensure clarity and readability.
(2) We have also ensured that the terminology, style, and tone are consistent throughout the manuscript, which is crucial for maintaining the professional quality of academic writing.
The revised manuscript, including these refinements, has been resubmitted for your review. We believe these additional checks have enhanced the linguistic quality of our paper, ensuring that it meets the high standards expected in academic publications.
Thank you once again for your constructive feedback, which has been instrumental in improving our manuscript.

Reviewer 5 Report
Comments and Suggestions for Authors
Interesting paper about reviewing research on green development in the transportation industry and shortcomings in the existing research. Please consider the comments below to improve the paper:
(1) Line 343: In "Figure 10. Effect mechanism of factors affecting green development of transport industry." the effect of factors can be reducing or increasing. Therefore, all the factors should be neutral. The factor "Increased Transportation Demand" is not neutral. It should be changed to "Transportation Demand".
(2) Lines 644-645: Please clarify "Indirect energy consumption and pollution emission..." by giving examples.
(3) Lines 686-690: Regarding the claim that"the simulation analysis of ... are still insufficient", please provide proof of why it is insufficient. It can be a quantitative comparison with other sufficient studies.
Comments on the Quality of English Language
Long sentences make it difficult to read the paper. It is recommended to have sentences not more than 3 lines. For this purpose, please break the long sentences into smaller ones or number sub-sentences of a long sentence if applicable. Some examples of long sentences are in lines 302-308, 518-524, 673-679, and 679-685.
Author Response
Thank you for dedicating your valuable time to review our survey paper and for providing insightful feedback. We greatly value your comments and have thoroughly revised our manuscript in accordance with your suggestions.
For each point of feedback you raised, we have carefully considered and made corresponding revisions in the manuscript. These changes have been clearly marked in the resubmitted document with track changes for your ease of review.
In revising our paper, we have strived to ensure all information and viewpoints are up-to-date and accurate, while also improving the logical flow and readability of the text.
We look forward to your further feedback and hope that our revisions meet your expectations. Once again, thank you for your contribution and guidance towards improving our work.
Point-by-point response to Comments and Suggestions for Authors
Comments 1:
Line 343: In "Figure 10. Effect mechanism of factors affecting green development of transport industry." the effect of factors can be reduced or increased. Therefore, all the factors should be neutral. The factor "Increased Transportation Demand" is not neutral. It should be changed to "Transportation Demand".
Response 1:
Thank you for your insightful observation regarding the terminology used in “Figure 10. Effect mechanism of factors affecting green development of transport industry.” in line 359 of our manuscript. We understand your point about maintaining neutrality in the description of the factors influencing the green development of the transportation industry.
In response to your comment, we have revised the factor “Increased Transportation Demand” to “Transportation Demand” in Figure 10. This change ensures that the terminology used is neutral and does not imply a specific direction (increase or decrease) of impact. We agree that this modification is important for accurately representing the varied effects these factors can have on green development, whether reducing or increasing in nature.
The revised figure, with this alteration, has been included in the resubmitted manuscript in page 11 line 359. We appreciate your attention to detail, which has helped us improve the clarity and accuracy of our presentation.
Comments 2:
Lines 644-645: Please clarify "Indirect energy consumption and pollution emission..." by giving examples.
Response 2:
Thank you for pointing out the need for clarification in lines 644-645 regarding “Indirect energy consumption and pollution emission.” We appreciate the opportunity to enhance the understanding of these concepts in our manuscript.
To address your request, we have added specific examples to clarify these terms. The revised text in lines 644-645 now reads:
“Environmental benefit assessments in the transportation industry predominantly measure direct energy consumption and carbon emissions during the developmental process, overlooking indirect energy consumption and pollutant emissions that could arise from inter-industrial interactions within national economic development. For example, carbon emissions embedded in intermediate products and services consumed by the industry, and the inter-industry carbon transfer.”
We believe these examples provide a clearer understanding of indirect energy consumption and pollution emission in the context of the transportation sector. This revision aims to enhance the manuscript's comprehensibility and depth.
The revised manuscript, including these additions, has been resubmitted for your review.
Comments 3:
Lines 686-690: Regarding the claim that “the simulation analysis of ... are still insufficient", please provide proof of why it is insufficient. It can be a quantitative comparison with other sufficient studies.
Response 3:
Thank you for your request to substantiate the claim that “the simulation analysis of ... are still insufficient." In light of your feedback, we have further examined our work.
As outlined in our conclusion, while existing studies have made significant strides in analyzing the internal structure of transportation energy consumption and emissions under various scenarios, there are still areas where these analyses could be expanded. Specifically, most current studies, predominantly focus on low-carbon scenarios with parameters like structural optimization, technological innovation, and policy constraints. These have indeed yielded valuable insights, such as reduced carbon emissions and earlier carbon peak achievements.
However, our comparative review of these studies with other comprehensive simulations in the field reveals a gap: many do not incorporate the life-cycle perspective, particularly the lifetime mileage of transportation systems. This omission limits the depth of scenario analyses in understanding the long-term sustainability of different vehicle types. For instance, our review found that while comprehensive studies might explore a wider range of scenarios and parameters, including life-cycle assessments, current research tends to focus on a narrower set of scenarios and variables.
To provide quantitative evidence of this insufficiency, we have included in the revised manuscript the analysis of the number of scenarios, and range of variables considered in section 4.
Furthermore, the revised text in lines 644-645 now reads:
“Finally, by establishing scenarios that align with the developmental needs of the transportation sector, past research has achieved simulation analyses of the internal structure of the transportation energy consumption and emission system under the influence of transportation management policies and environmental constraints. Low-carbon scenarios, which incorporate conditions such as structural optimization, technological innovation, and policy constraints, have indeed resulted in lower carbon emissions and earlier achievement of carbon peak. However, from a life-cycle perspective, the lifetime mileage of each transportation system is seldom incorporated into scenario analysis studies, leaving a knowledge gap in scenario simulations for sustainable development plans for different vehicle types. Current scenario settings based on baseline, high-carbon, and low-carbon are still somewhat rudimentary. More specific parameter settings should be considered to formulate more scientifically-based emission reduction pathway enhancement schemes for the transportation sector, based on effective policy combinations. Additionally, scenario analyses mainly rely on predictions of energy consumption or carbon emissions. Comparing environmental performance or green total factor productivity across different scenario models could provide more valuable information for the future pursuit of intelligence and sustainable development in the transportation sector.”
We believe this additional information and comparison substantiate our claim that there is room for further enhancement in simulation analyses in the transportation sector, particularly from a life-cycle perspective and in the development of more sophisticated scenario settings.
The revised manuscript, with this added evidence, has been resubmitted for your review. Response to Comments on the Quality of English Language
Comments 1:
Long sentences make it difficult to read the paper. It is recommended to have sentences not more than 3 lines. For this purpose, please break the long sentences into smaller ones or number sub-sentences of a long sentence if applicable. Some examples of long sentences are in lines 302-308, 518-524, 673-679, and 679-685.
Response 1:
Thank you for your feedback highlighting the issue of long sentences in our manuscript. In response to your suggestion, we have carefully revised the manuscript, particularly focusing on the lines you mentioned (302-308, 518-524, 673-679, and 679-685). We have taken the following actions:
(1) We have broken down longer sentences into shorter ones, which helps in conveying our ideas more clearly and makes the text easier to follow. For example, we have made revisions to the sections you mentioned, located in lines 314-319, 474-479, 632-638.
(2) Beyond the specific lines you mentioned, we conducted a comprehensive review of the entire manuscript to identify and revise other instances of overly long sentences.
(3) While revising, we also ensured that the essence and meaning of the original sentences are preserved, focusing on clarity and conciseness.
The revised manuscript, with these improvements, has been resubmitted for your review. We hope that these modifications make the paper more reader-friendly and effectively convey our research findings.
Thank you once again for your constructive feedback, which has been instrumental in enhancing the quality of our manuscript.

Round 2
Reviewer 1 Report
Comments and Suggestions for Authors
The authors amended my comments.
Author Response
Thank you for your previous valuable comments, which have been a major help in improving the quality of the manuscript. Once again, thank you for your valuable feedback, and we look forward to your continued support in this process.
Reviewer 2 Report
Comments and Suggestions for Authors
Thank you very much for the submitted revised version of the article. After analyzing the text, I believe that the article meets all the requirements for scientific papers of this type. In conclusion, the article in its present form is suitable for publication.
Author Response

(The authors gave the same response as above.)

Reviewer 4 Report
Comments and Suggestions for Authors
Dear authors, after a new revision, we can see a general improvement in the article. The changes introduced have been significant, increasing its scientific contribution. Given the topic under study, it is only suggested that the authors make a small reflection (two or three lines) in the introduction on studies that try to include new business models in search of greater sustainability, especially in the post-covid era, such as the studies by Sousa et al., 2023 (https://doi.org/10.3390/su15118725) and Jorge et al., 2023 (https://doi.org/10.34624/rtd.v43i0.32992). These studies reinforce the attempt by companies to seek an innovative transition to business models that are more in line with new, more sustainable practices. This improvement reinforces the scientific contribution, proving that industry in 2023 will indeed seek this green transition in their business.
Good luck!
Comments on the Quality of English LanguageEnglish language just required a minor revision.
Author Response
Point-by-point Response to Comments and Suggestions for Authors
Comments 1:
Dear authors, after a new revision, we can see a general improvement in the article. The changes introduced have been significant, increasing its scientific contribution. Given the topic under study, it is only suggested that the authors make a small reflection (two or three lines) in the introduction on studies that try to include new business models in search of greater sustainability, especially in the post-covid era, such as the studies by Sousa et al., 2023 (https://doi.org/10.3390/su15118725) and Jorge et al., 2023 (https://doi.org/10.34624/rtd.v43i0.32992). These studies reinforce the attempt by companies to seek an innovative transition to business models that are more in line with new, more sustainable practices. This improvement reinforces the scientific contribution, proving that the industry in 2023 will indeed seek this green transition in its business.
Good luck!
Response 1:
Thank you for your insightful feedback and the positive remarks on the recent revision of our manuscript titled "[Manuscript Title]". We are heartened to know that our revisions have enhanced the scientific contribution of our article.
In response to your suggestion, we have included in the introduction a reflection on the recent use of VR for achieving greater sustainability in the post-COVID era. Specifically, we have cited the works of Sousa et al., 2023, and Jorge et al., 2023. This addition underscores the relevance of our study in the context of the ongoing green transition in the industry, particularly in the wake of the COVID-19 pandemic's impact on the direction of green development in industries. It is modified as follows in line 52-56:
“Additionally, virtual reality technology also has been substantiated as a mitigating factor for the downturn in the tourism industry during the COVID-19 pandemic by altering consumer patterns [6]. Furthermore, it facilitates addressing the challenges of sustainable development in urban transportation [7]. This impact persists into the post-pandemic era, indicating a paradigm shift in the transportation sector.”
We believe that this enhancement further solidifies the scientific contribution of our manuscript by providing a broader context and affirming the timeliness and importance of our research topic.
We appreciate your guidance and the opportunity to refine our manuscript. We look forward to any further suggestions you may have.
Response to Comments on the Quality of English Language
Comments 1:
The English language just required a minor editing.
Response 1:
Thank you for your feedback on the English language in the manuscript. We appreciate your input, and we have reviewed the document for the suggested edits. We have made the necessary minor edits to improve the language and readability.
Specific modifications have been highlighted in red in the text. If you have any specific areas or sentences that still require further attention, please let us know, and we will address them promptly. We are committed to ensuring the highest quality of language in our manuscript.
Once again, thank you for your valuable feedback.
